# Multiple kinesins induce tension for smooth cargo transport

**Marco Tjioe[1,2,3†], Saurabh Shukla[2,4], Rohit Vaidya[1,2], Alice Troitskaia[1], Carol S Bookwalter[5], Kathleen M Trybus[5], Yann R Chemla[1,2,3], Paul R Selvin[1,2,3]\***

[1]Center for Biophysics and Quantitative Biology, University of Illinois at Urbana-Champaign, Urbana, United States; [2]Center for the Physics of Living Cells, University of Illinois at Urbana-Champaign, Urbana, United States; [3]Department of Physics, University of Illinois at Urbana-Champaign, Urbana, United States; [4]Department of Chemical and Biomolecular Engineering, University of Illinois at Urbana-Champaign, Urbana, United States; [5]Department of Molecular Physiology and Biophysics, University of Vermont, Burlington, United States

**Abstract** How cargoes move within a crowded cell—over long distances and at speeds nearly the same as when moving on unimpeded pathway—has long been mysterious. Through an in vitro force-gliding assay, which involves measuring nanometer displacement and piconewtons of force, we show that multiple mammalian kinesin-1 (from 2 to 8) communicate in a team by inducing tension (up to 4 pN) on the cargo. Kinesins adopt two distinct states, with one-third slowing down the microtubule and two-thirds speeding it up. Resisting kinesins tend to come off more rapidly than, and speed up when pulled by driving kinesins, implying an asymmetric tug-of-war. Furthermore, kinesins dynamically interact to overcome roadblocks, occasionally combining their forces. Consequently, multiple kinesins acting as a team may play a significant role in facilitating smooth cargo motion in a dense environment. This is one of few cases in which single molecule behavior can be connected to ensemble behavior of multiple motors.

**\*For correspondence:**
selvin@illinois.edu

**Present address:** [†]Element Biosciences, San Diego, United States

## Introduction

Kinesin is part of a cytoskeletal motor family that moves cellular cargoes primarily to the cell periphery (microtubule plus end). It is important in key cellular processes like cell division and signaling (*Gross et al., 2002*). It is also implicated in several neurological disorders (*McLaughlin et al., 2016*). Due to advances in single molecule microscopy and force measurement techniques, transport properties by a single kinesin are well understood (*Veigel and Schmidt, 2011*). For example, kinesin-1, the prototypical kinesin, moves 8.4 nm per ATP consumed, in a hand-over-hand motion, walking about 100 steps before detaching and traveling at a speed of ~0.77 μm/sec in vitro (*Cai et al., 2007*), and equal- or higher speed in vivo (*Block et al., 1990*; *Stamer et al., 2002*; *Yildiz et al., 2004*). A single kinesin also exerts up to ~6 pN force (*Svoboda and Block, 1994*), and importantly, has an asymmetric run-length and velocity with regard to the direction of force on the microtubule (*Figure 1A*) (*Coppin et al., 2002*; *Milic et al., 2014*).

However, a cell is extremely dense, filled with proteins (~300 mg/ml), and only 60–80% water volume (*Albe et al., 1990*), resulting in a high viscosity and elastic modulus (*Berret, 2016*). Despite this, large cargos move at virtually the same rate as in a water-based in-vitro environment (*Furuta et al., 2013*; *Howard et al., 1989*). How is this possible? We argue that it is due to the action of multiple motors acting on a single cargo.

Multiple motor transport is important in cellular trafficking (*Blehm et al., 2013*; *Gross et al., 2002*; *Hendricks et al., 2012*; *Holzbaur and Goldman, 2010*). Electron microscopy shows that ~1 to 7 motors are bound to cellular cargoes (*Gross et al., 2007*). Motors are inter-dependent

**eLife digest** The inside of a cell is a crowded space, full of proteins and other molecules. Yet, the molecular motors that transport some of those molecules within the cell move at the same speed as they would in pure water – about one micrometer per second. How the molecular motors could achieve such speeds in crowded cells was unclear. Nevertheless, Tjioe et al. suspected that the answer might be related to how multiple motors work together.

Molecular motors move by walking along filaments inside the cell and pulling their cargo from one location to another. Other molecules that bind to the filaments should, in theory, act like "roadblocks" and impede the movement of the cargo. Tjioe et al. studied a motor protein called kinesin, which walks on filaments called microtubules. But instead of looking at these motors moving along microtubules inside a cell, Tjioe et al. used a simpler system where the cell was eliminated, and all parts were purified. Specifically, Tjioe et al. tethered purified motors to a piece of glass and then observed them under an extremely accurate microscope as they moved free-floating, fluorescently labelled microtubules. The microtubules, in this scenario, were acting like cargoes, where many kinesins could bind. Each kinesin motor also had a small chemical tag that could emit light. By following the movement of the lights, it was possible to calculate what each kinesin was doing and how the cargo moved.

When more than one kinesin molecule was acting, the tension and speed of one kinesin affected the movement of the others. In any group of kinesins, about two-thirds of kinesin pulled the cargo, and unexpectedly, about one-third tended to resist and slow the cargo. These latter kinesins were moved along with the group without actually driving the cargo. These resisting kinesins did come off more rapidly than the driving kinesins, meaning the cargo should be able to quickly bypass roadblocks. This would help to keep the whole group travelling in the right direction at a steady pace.

---

(*Gross et al., 2002*), such that the impairment of one motor type (e.g. dynein) causes severe impairment in the other (e.g. kinesin) (*Gross et al., 2002*). Run length and stall forces are typically greater for multiple vs single motor, although not necessarily in proportion (*Holzbaur and Goldman, 2010*). Theoretical studies predict tension between multiple motors carrying the same cargo (*Arpağ et al., 2014*). This tension, we argue, allows for their ability to efficiently bypass roadblocks, which has yet to be shown experimentally.

To understand multiple motor transport, it is crucial to probe the force and motion of each kinesin, plus that of the cargo motion. However, no current single molecule assay can achieve this. For example, atomic force microscopy (AFM) allows recording of only single motors walking (*Kodera and Ando, 2018*); current fluorescence and optical trap assays can measure overall forces and positions of multiple motors, but not every single motor (*Derr et al., 2012*; *Jamison et al., 2010*). The inability to probe all molecular motors may have led to the differing conclusions on the effect of motor number on run length (*Block et al., 1990*; *Derr et al., 2012*; *Efremov et al., 2014*; *Shubeita et al., 2008*; *Vershinin et al., 2007*). Two in vitro assays come close to probing the behavior of all participating kinesins. One uses a programmable DNA origami with up to seven kinesins attached (*Derr et al., 2012*). Another uses immobilized GFP-labeled kinesins and tracks gliding microtubules with quantum dots attached (*Leduc et al., 2007*). These studies accurately count the number of kinesins on cargo, but not their behavior.

One pertinent question is whether single-motor asymmetry plays a role in multi-motor transport. The Block group found an asymmetry when studying the *run length* of single kinesin as a function of force in an optical trap (*Milic et al., 2014*) (*Figure 1A*). This asymmetry is between driving kinesin, which experiences force opposite to its walking direction, and resisting kinesin, which experiences force in the same direction as its movement. They found that the run length is greater for driving kinesin (50 nm at −6 pN rising rapidly to 1100 nm at 0 pN) vs. resisting kinesin (~100 nm at 2 pN slowly falling to 10 nm at 20 pN) (*Figure 1A*) (*Milic et al., 2014*). Similarly, the Vale group saw kinesin *velocity* to be slower for driving-kinesin than resisting-kinesin (*Figure 1B*) (*Coppin et al., 2002*). Whether this run-length and velocity asymmetry occurs when multiple driving and resisting kinesins

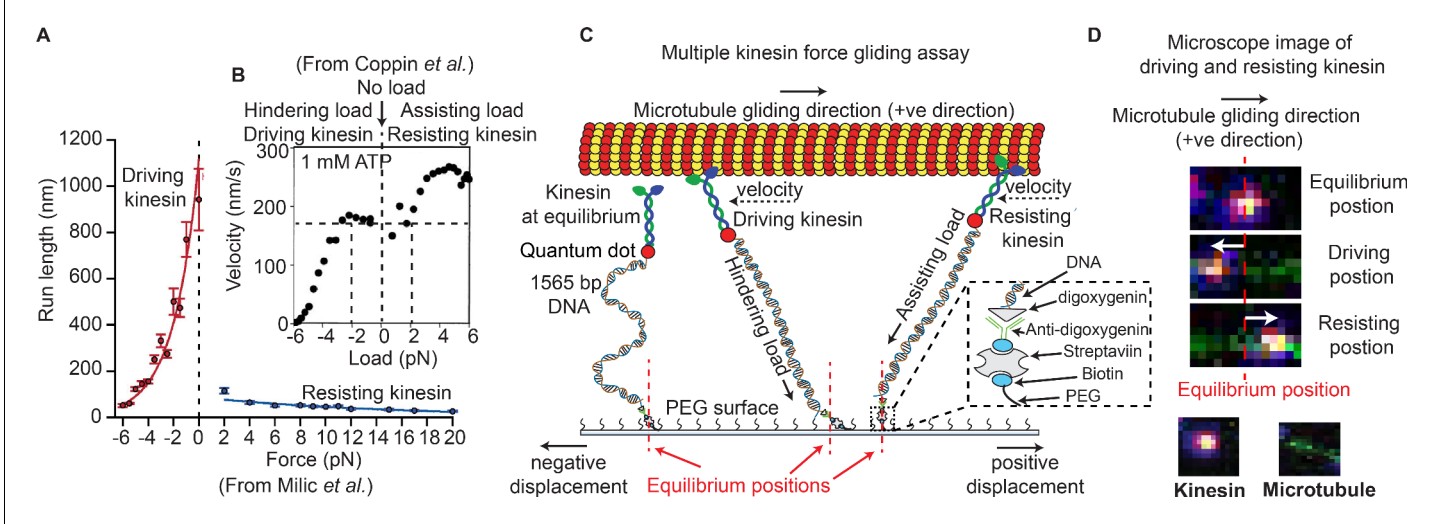

**Figure 1.** Force gliding assay to study multiple kinesins transporting a cargo. (A) Kinesin run-length is asymmetric in response to load as measured in an optical trap (graphs reproduced from *Milic et al., 2014*). (B) *Coppin et al. (2002)* observed an asymmetric velocity response to oppositely applied load. Driving kinesin velocity is slower (0 nm/s to ~170 nm/s between -6 pN to -2 pN and then constant at 0 pN load) than resisting kinesin (180 nm/s to 280 nm/s between 0 pN to 6 pN load) when pulled by loads in opposite directions. The graph is obtained from 100 kinesin runs. (C) Schematic of force gliding assay to study behavior of multiple kinesin motors transporting a single microtubule as a cargo. Kinesin-QD is attached to the surface with a 1565-bp long dsDNA linker that allows the detection of kinesin-QD motion as it drives or resists the cargo (cargo moves towards the right); the equilibrium position of kinesin is also shown. The displacement of resisting kinesin is in the same direction as the microtubule velocity (positive displacement by convention) and the position of driving kinesin is in the opposite direction to the microtubule velocity (negative displacement). Using the extensible worm-like chain model (eWLC), forces on each kinesin could be estimated. (D) Microscope images of driving and resisting kinesins are shown. Kinesin is at first in its equilibrium position (top image). When kinesin assumes a driving position, it is displaced to the left of its equilibrium position (middle image). When kinesin becomes resisting, it moves to the right, in the microtubule velocity direction (bottom image). The equilibrium position is depicted by the red dotted line.

are carrying a single cargo was not determined, and whether inter-kinesin interactions cause this, will be described here.

To overcome current experimental limitations, a number of authors have done simulations which provided insights into multiple-motor behavior. One simulation showed that the asymmetric property of single kinesin under load leads to an average of one-third of kinesin's resisting in multi-motor situation (*Arpağ et al., 2014*). In addition, the tension was between 0 to 15 pN between the kinesins carrying the cargo (*Arpağ et al., 2014*). Other simulations showed that, on a single cargo being driven by multiple motors, force-dependent detachment of motors (*Arpağ et al., 2014*), particularly the resisting motors (*Nelson et al., 2014*), is important for cargo speed. However, the accuracy of simulations heavily depended on the particular models used (*Arpağ et al., 2014*; *Kunwar and Mogilner, 2010*; *Kunwar et al., 2008*; *Xu et al., 2013*).

Another question is how well multiple motors cooperate together. A number of experimental studies found negative cooperativity of kinesin (*Furuta et al., 2013*; *Jamison et al., 2010*). Negative cooperativity, in this case, means that cargo transport does not fully benefit from having two motors present due to a certain inter-motor inhibition that reduces the motor-filament binding energies in the system. Motor run length and average detachment forces would therefore not increase as dramatically as expected in cases without this interference (*Jamison et al., 2010*). However, exactly how the on- and off-rates of each kinesin (e.g. run length and binding duration) change to give rise to negative cooperativity is unknown.

Negative cooperativity does not mean that additive forces of two or more kinesins can never occur. It can happen occasionally, and can be observed in optical trap assays (*Jamison et al., 2010*; *Vershinin et al., 2007*). How relevant additive forces are to cellular transport is unclear. It is possible

that additive forces help kinesin get unstuck upon roadblock encounter, but this has not been shown. Another possibility is that additive forces can detach kinesin(s) from the cargo, which has also not been shown, but may facilitate cargo transport in a crowded cell (*Conway et al., 2012*). It is possible that additive forces happen only transiently, but, nevertheless, may be important to bypass roadblocks.

In this work, we have developed an in-vitro assay, which we call a force-gliding assay, details of which are discussed below. It allows direct observation of individual kinesin-1's motion, velocities and forces, acting as a *dynamical* team of multiple kinesin motors (1- ~ 8), transporting a common cargo (microtubule), whose position and velocity can be measured. We can directly observe the attachment and detachment of individual motors from the microtubule, and find that kinesin exists in two distinct states, one driving the microtubule, the other resisting. We observe an asymmetric run length and velocity response to load of driving-kinesins and resisting-kinesins, indicating a non-zero tension between kinesins carrying the same cargo. This leads to one-third of kinesins always resisting. We further show that multiple kinesins exhibit negative cooperativity through decreasing run-length of individual kinesins as more kinesins participate in transport. Lastly, multiple kinesins can combine forces that help in overcoming the roadblocks.

We conclude that an asymmetric tug-of-war, with negative cooperativity and additive forces, defines collective kinesin transport and can potentially help with uninterrupted cargo transport inside the cell. We note that our assay measures *dynamic* interactions between *multiple* kinesins and a cargo. The ability to do get dynamic interactions may be crucial because it may be only transient interaction which are necessary to bypass roadblocks. Whether the tension embedded within an in vivo system, and whether other molecular motors such as dynein and myosin have a similar cooperative behavior, remains to be seen.

## Results

### Force-gliding assay allows simultaneous interrogation of multiple kinesin motors

We developed a modified microtubule gliding assay, called a force-gliding assay, capable of measuring the direction and the magnitude of force exerted by individual kinesin-1 on the microtubule cargo in real-time (*Figure 1C*). The assay can also be used to estimate the attachment-detachment dynamics of each kinesin. Each kinesin was labeled with a quantum dot (QD) at a ratio of 1:1, and attached by a 1565-base pair dsDNA molecule, which acts like a non-linear spring, to a non-stick polyethylene-glycol (PEG) coverslip. Issues of multiple DNA-binding to a QD and non-fluorescent QD have been minimized (see Appendix 1). A fluorescently-labeled microtubule (shown moving to the right in *Figure 1C*) serves as the cargo and is moved by kinesins at saturating (1 mM) ATP conditions. The points at which kinesins are attached to the glass coverslip through the dsDNA are defined as the 'equilibrium positions'. The position of each kinesin could then be monitored with nanometer accuracy via a tracking algorithm similar to Fluorescence Imaging with One Nanometer Accuracy (FIONA) (*Yildiz and Selvin, 2005*) (see Materials and methods). Single particle tracking was possible because the kinesins were placed at a distance greater than the diffraction limit apart, allowing many kinesins to be monitored individually, but simultaneously. The force acting on the kinesins can be estimated from the DNA extensions using the extensible Worm Like Chain (eWLC) (*Lee and Thirumalai, 2004*).

Upon tracking the positions of kinesins during cargo transport, we found that kinesins were in one of two possible states, shown in *Figure 1C*. One state speeds up the microtubule—called the 'driving kinesin', where the kinesin pushes the microtubule in the direction of the microtubule gliding (*right* in *Figure 1C*), causing the kinesin to be displaced in the opposite direction of microtubule gliding (*left* of kinesin's equilibrium position in *Figure 1C*). The other state, called the 'resisting kinesin', slows the microtubule down, resulting in the kinesin being displaced in the direction of microtubule gliding (*right* of kinesin's equilibrium position in *Figure 1C*). Therefore, we define signs such that the resisting kinesins have positive displacements from their equilibrium position and the driving kinesins have negative displacements. Both fluorescent signals from the QD on the kinesin and the organic fluorophores on the microtubule allowed several minutes-long recording. Using the force-

gliding assay, we estimated the displacement of each kinesin with respect to their equilibrium positions, as well as measure the corresponding microtubule velocity in real time.

## Driving and resisting kinesins can dynamically switch roles

To investigate the dynamics of multiple kinesins in detail, we tracked the motion of each microtubule when it was being moved by multiple kinesins (see Materials and methods). *Figure 2* is an example of one of these cases where a microtubule (green) is transported by three kinesins, labeled #1, 2 and 3 (see *Video 1*). *Figure 2A* shows the microscope images at different time points during the microtubule transport. The top image of *Figure 2A* (t = 0 s) shows that three kinesins are at their equilibrium position (marked by yellow arrows, when they are not transporting any microtubule). When the microtubule transport begins at t = 1.4 s, kinesins start dynamically fluctuating around their respective equilibrium positions. We observed that individual kinesin while transporting the microtubule will be in either the driving or resisting states, and can switch between them. For example, kinesin three is in the resisting state at t = 5.2 s and transitions to the driving state at t = 14 s (*Figure 2A*).

*Figure 2B* shows the microtubule velocity and kinesin displacement analysis of the same three kinesins. xc shows the kymograph of the microtubule. We calculated the microtubule velocity, plotted in *Figure 2B2*, by tracing the microtubule kymographs. We tracked the position of individual kinesins, plotted their displacements and correlated them to the microtubule velocity in *Figure 2B3*. The dashed horizontal black lines in the kinesin displacement plots represent the equilibrium position of each kinesin. Dashed blue boxes, expanded at the bottom of *Figure 2B4 and B5*, show the resisting and driving displacements above (positive) and below (negative) the equilibrium lines in detail. Four time points (same as in *Figure 2A*) are depicted by dashed yellow vertical lines. At t = 0 s, all kinesins are at their equilibrium position. At 1.4 s, a microtubule appears; kinesin #1 and #3 are still at their equilibrium positions, while kinesin #2 rapidly starts driving the microtubule—therefore it has negative displacement. Due to kinesin #2 assuming a driving role, the microtubule velocity increases to ~850 nm/sec. At 5.2 s, kinesin #3 is resisting while kinesin #1 and #2 are driving, maintaining the velocity of microtubule at ~850 nm/sec. At 14 s, kinesin #3 is driving the microtubule while kinesin #2 is resisting the motion, resulting in a ~ 600 nm/sec microtubule velocity. This shows that kinesins dynamically switch their roles, which affects the cargo velocity. The fact that kinesins, while working together, can switch states from driving to resisting and vice versa, can be made quantitative. We measured many such transitions (N = 685), similar to that shown in *Figure 2A and B*. The average transition rate of a single kinesin is 4.9 transitions/min (*Figure 2C*). The majority (52%) of these transitions are drive-to-drive transitions, during which kinesin drives, then return to equilibrium while still being attached to or after detaching from the microtubule, and then drives again. The drive-to-resist transitions and resist-to-drive percentage contribute 16% of all transitions each. Together they make up approximately one-third of all transitions. This shows the highly dynamic nature of kinesin working in a team. Rather than one kinesin taking the lead all the time, kinesins are constantly changing roles between driving and resisting.

## Run length asymmetry leads to a 0–4 pN tension between the kinesins and a constant fraction of kinesin driving or resisting, independent of the amount of kinesin present

Next, we ask if there is a difference in run length and duration between kinesins in the driving and the resisting states, as observed in the single kinesin case of *Milic et al. (2014)*. From the Milic et al. data (*Figure 1A*), we can say that without tension, that is with F = 0, the forward and backward run lengths are equal to each other (~1 μm). Run length asymmetry for multiple motor transport will only be observed if there is non-zero tension between driving and resisting kinesins carrying the same cargo, as shown in *Figure 1A* for F ≠ 0. With multiple kinesins, *Figure 3A* shows that driving kinesin stays attached to microtubule for an average of 3.0 ± 0.21 s, compared to 2.15 ± 0.15 s for resisting kinesin. *Figure 3B* shows that the average run length of driving-kinesin is 2.31 ± 0.18 μm, compared to 1.42 ± 0.09 μm for resisting kinesin. Our observation of asymmetry in run length and duration suggests there is tension between driving and resisting kinesins with multiple motors driving a single (microtubule) cargo.

To quantify the tension when multiple kinesins carry the same cargo, we used a semi-quantitative method to estimate the force (or tension) on the kinesin based on the extensible WLC model

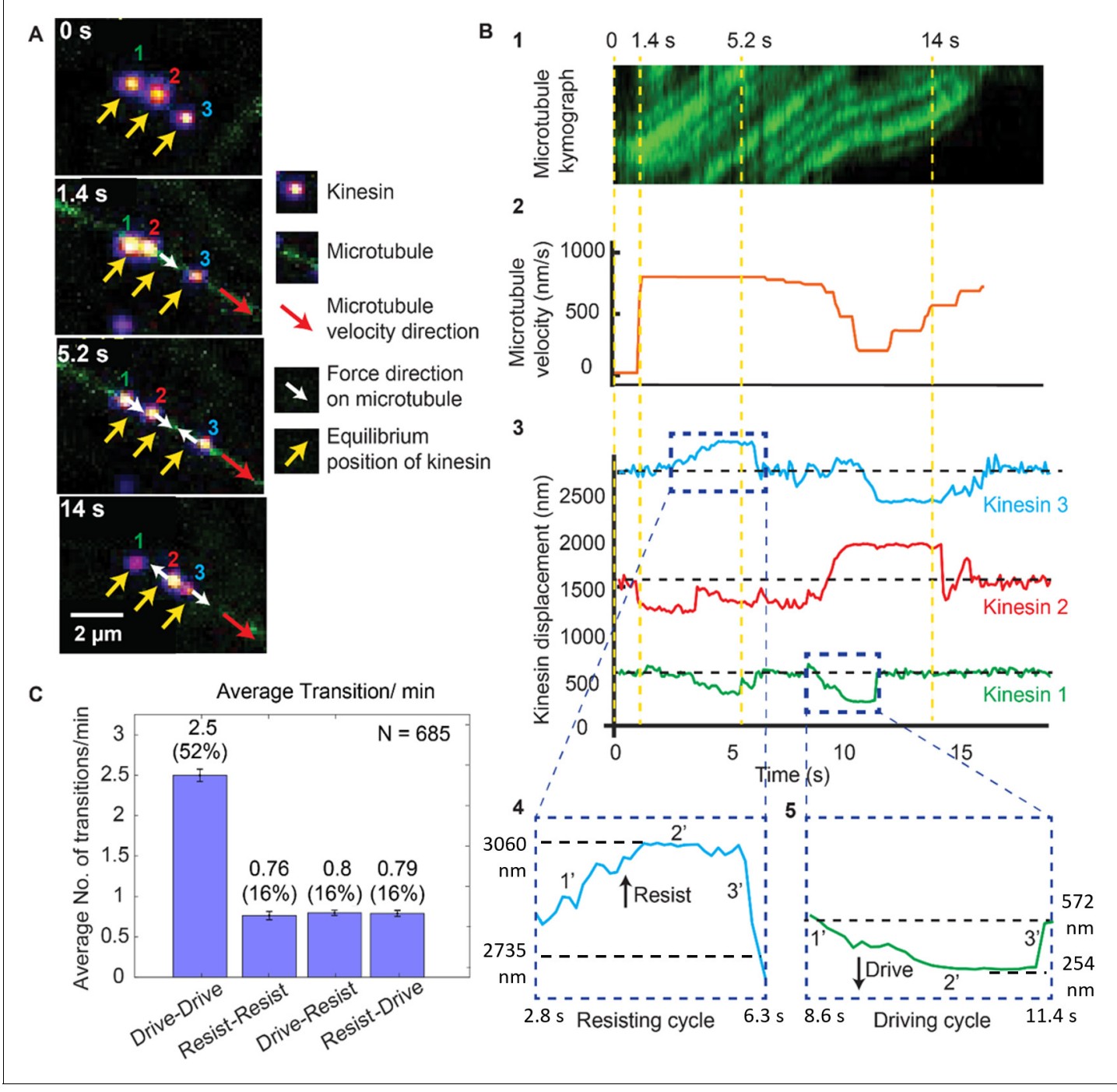

**Figure 2.** Dynamics of driving and resisting kinesins and their effect on microtubule velocity. (**A**) One case of a force gliding assay is presented where three kinesins labeled with 705-nm QDs, marked 1, 2 and 3, move a labeled microtubule (green). Raw images at 0, 1.4, 5.2 and 14 sec taken every 0.1 sec (see **Video 1**) are shown. Yellow arrows show equilibrium kinesin positions. White arrows point along the direction of the force generated by kinesins on microtubule as they are displaced from their equilibrium positions. Red arrow shows the direction of microtubule movement. (**B**) Plots showing the microtubule kymograph and velocity, and kinesin positions over time. Time points 0, 1.4, 5.2, and 14 sec (corresponding to **A**) are marked with yellow vertical lines. Microtubule velocity (middle panel) starts from 0 nm/s at t = 0 sec, and increases to ~850 nm/s at frame 1.4 sec, when kinesin #2 starts driving (negative kinesin displacement). At 5.2 sec, kinesin #1 joins kinesin #2 to drive the microtubule, while kinesin #3 starts resisting. At 14 sec, kinesin #2 is resisting, kinesin #3 is driving and kinesin #1 is in equilibrium position. The fluctuation in kinesin displacement results from both Brownian motion and tracking error. Driving and resisting cycles are zoomed in for detail. Points 1', 2' and 3' are transition regions. At positive/negative slope of 1' region, kinesin slow down (for resisting cycle) or speed up (for driving cycle) with respect to microtubule. At the zero slope of the 2'

*Figure 2 continued on next page*

*Figure 2 continued*

region (plateau region), kinesin's speed is the same as that of microtubule as DNA is fully stretched. At the 3' region, kinesin returns to equilibrium. (**C**) The average number of transitions per minute. The major transition is from driving to driving.

The online version of this article includes the following figure supplement(s) for figure 2:

**Figure supplement 1.** Force vs DNA extension of 1565 base DNA calculated using eWLC.

(*Lee and Thirumalai, 2004*). Using a persistence length of 50 nm, a dsDNA contour length of 532 nm and a distance-offset of 20 nm to account for the size of QD, proteins and other attachment agents, we obtained the average force as a function of time on kinesins that is shown in *Figure 3C*. A total of 1478 driving kinesins and 573 resisting kinesins was used. Before 0.8 s, which is the time for dsDNA to stretch near its contour length, assuming a kinesin velocity of 800 nm/s, the tension is on average less than ~0.4 pN. After 0.8 s, the tension increases and stays between below ~4 pN until kinesin detaches. Therefore, we estimate the tension between kinesins to be between 0 and 4 pN.

Because of the uncertainty in DNA extension (due to uncertainty in distance offset,~0–20 nm, and equilibrium point determination,~0–40 nm: see Appendix 2), there is significant uncertainty in the force results—from sub-pN up to tens of pN—depending on the DNA extension (Sup. *Figure 1*). We, therefore, sought to verify our force results with the published literature. In particular, Milic et al., found that the relevant forces are 0 to −4 pN for driving kinesins and from 0 to 4 pN for resisting kinesin. Outside of this range (F< −4 pN or F > 4 pN), the run length difference is small between driving and resisting kinesin. (Note that Milic et al. refer to the load being carried by the kinesin, where the hindering load corresponds to the driving kinesin and the assisting load to the resisting kinesin: see *Figure 1B*). Consequently, our results of 0–4 pN tension between driving and resisting kinesins are in semi-quantitative agreement with the result of Milic et al.

Knowing that run length in multiple kinesin transport is asymmetric, we then asked how this affects the fraction of driving and resisting kinesins. We expected that the longer run length of driving kinesin would result in more kinesins being in the driving state than the resisting state. Indeed, we found that there is a constant fraction—about 2/3—of kinesins which are driving. This means that about 1/3 of the kinesins are resisting, regardless of the total number of kinesins (from 1 to 8) attached to the microtubule (*Figure 3D*). The fact that about 1/3 of the kinesins are in the resisting mode has significant implications, to be discussed in the Discussion section.

## Velocity asymmetry enables kinesins to match the cargo speed in multiple motor transport

Next, we asked if the velocity of the driving and resisting kinesins change differently when pulled in opposite directions, that is is there a velocity asymmetry? For a single kinesin, *Coppin et al. (2002)* found that the answer was yes: directional loads—which, in their case, are imposed by pulling the kinesin forward or backwards on a stationary microtubule—will slow down driving kinesin, and speed up resisting kinesin. There is a force range around zero, from −2 pN to +2 pN, where there is no change, that is there is no asymmetry (*Figure 1B*).

Such an asymmetric effect on the velocity has *not* been shown for multiple kinesins. In multiple motor case, asymmetry can arise because a kinesin can be in the driving mode or the resisting mode *and* apply tension (force). We find that for multiple kinesins, the driving kinesins slow down to match the cargo (microtubule) rate, and the resisting kinesins speed up to match the cargo (microtubule) rate. This speeding up or slowing down can be seen in the plateaus for the kinesin displacement graphs. Two examples of the plateaus are shown as the 2' region in the resisting kinesin trace (*Figure 2B4*) and driving kinesin trace (*Figure 2B5*). What is happening is because there is asymmetry in the tension direction

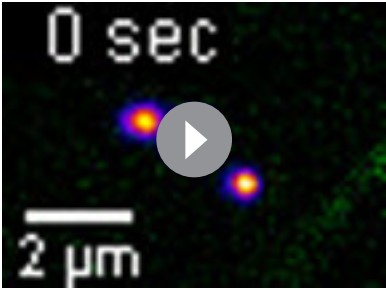

**Video 1.** Three kinesins moving a microtubule.
https://elifesciences.org/articles/50974#video1

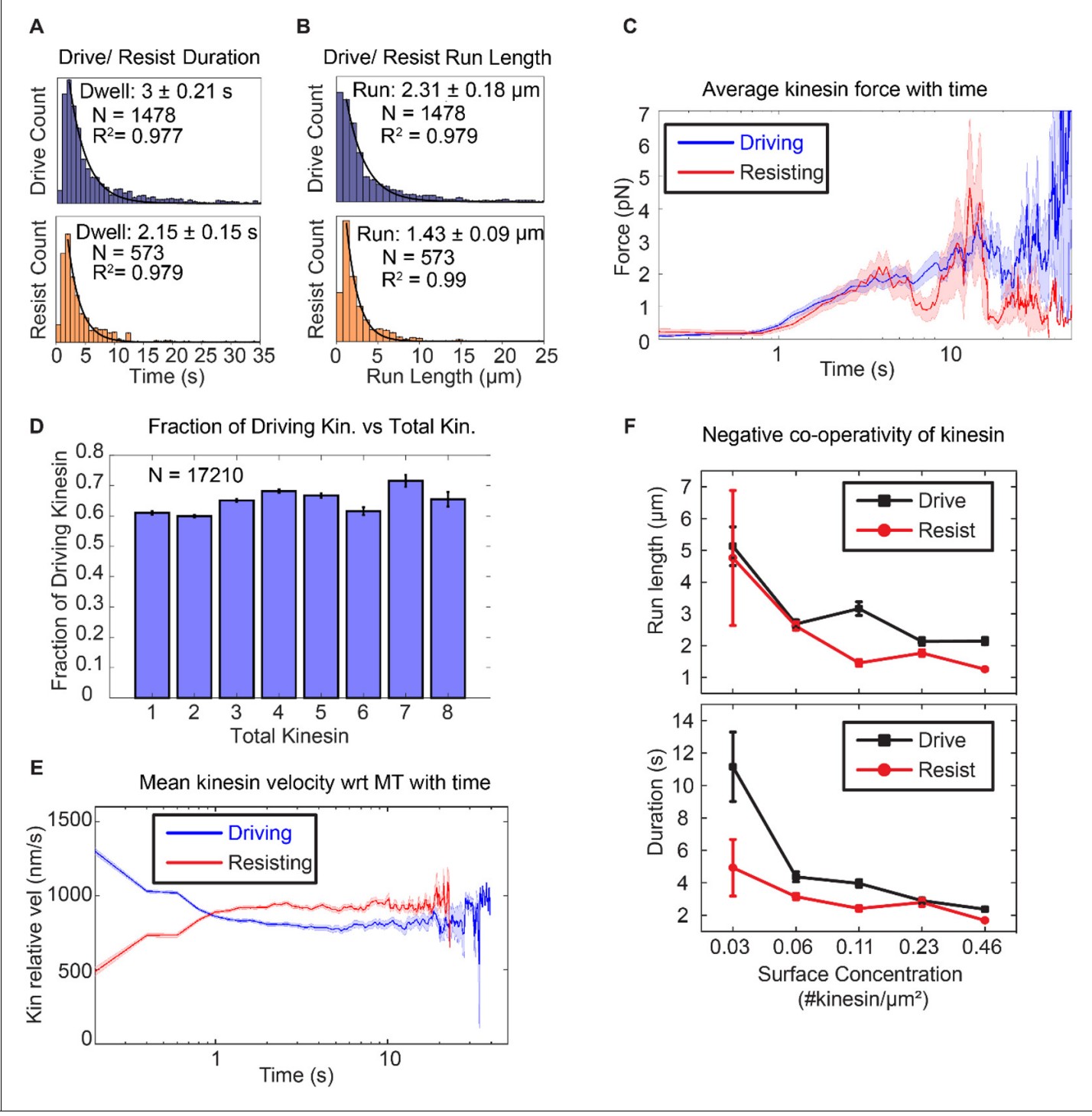

**Figure 3.** Asymmetric response of kinesin. (**A**) and (**B**) Combining all the lifetimes (durations) and run lengths from 1478 driving and 573 resisting kinesins, driving kinesin stays attached approximately 40% longer and walks approximately 62% further than the resisting kinesin, showing that the resisting kinesin tends to detach more readily than the driving kinesin. (**C**) The average force exerted by driving/resisting kinesin over time increases from less than ~1 pN when it first binds (t < ~0.8 s) to ~3-4 pN at ~5 sec. Light blue and light red shadings are the standard error of the mean. A total of 1478 driving kinesins and 573 resisting kinesins were used to generate this graph. (**D**) The fraction of driving kinesins remains approximately constant at around ~65% as the number of kinesins attached to a microtubule increases. (**E**) The average velocity of kinesin relative to the microtubule is plotted for 1478 driving kinesins and 573 resisting kinesins (detailed derivation in *Figure 4—figure supplement 4*). Driving kinesins start at a higher relative velocity (~1300 nm/s) than resisting kinesins (~500 nm/s). This is due to natural variation in kinesin velocity, as at the start of motion kinesin feels negligible force from other kinesins (DNA is not stretched). Since driving and resisting kinesins transport the same microtubule, they eventually

*Figure 3 continued on next page*

*Figure 3 continued*

acquire the same speed. (**F**) The run length and duration of driving and resisting kinesins decrease as the kinesin surface concentration is increased from 0.03 to 0.46 kinesin/µm² ; this shows that as more kinesins are involved in moving the microtubule, each kinesin stays attached to and walks on the microtubule for a shorter duration and run length. A total of 36, 215, 169, 440 and 618 driving kinesins and 13, 83, 86, 149 and 242 resisting kinesins are used to generate the points at 0.03 to 0.46 kinesin/ µm² surface concentrations.

The online version of this article includes the following figure supplement(s) for figure 3:

**Figure supplement 1.** Effect of increasing kinesin surface concentration to the total number driving and resisting kinesin.

between the two types of kinesins, there is a concomitant asymmetric change in velocity (see also Appendix 7).

To test the prevalence of velocity asymmetry in multiple kinesin transport, we computed the average velocity of 1478 driving kinesins and 573 resisting kinesins relative to the microtubule, summarized in the result of *Figure 3E*. We observe slowing down of driving kinesins and speeding up of resisting kinesins, confirming velocity asymmetry in bulk kinesin behavior. On average, the driving kinesins start at a velocity of ~1200 nm/s and resisting kinesins at ~500 nm/s relative to microtubule (=absolute velocity of the microtubule minus absolute velocity of the kinesin) (*Figure 1A*). Driving kinesins slow down and resisting kinesins speed up to ~830 nm/s, which is the average microtubule velocity (taken from the velocity statistics discussed later in *Figure 4C*). Note that at ~1 s, the driving kinesin velocity intersects with resisting kinesin velocity, which is due to kinesin detachment, as explained in Appendix 3 and *Figure 4—figure supplement 4*). Overall, we show that velocity asymmetry can be observed in individual driving and resisting traces that are involved in multiple kinesin transport. Such velocity asymmetry leads to resisting and driving kinesins speeding up or slowing down, respectively, to match the velocity to the cargo speed.

## Increasing the number of kinesins reduces their run length

How well do kinesins work together? Some studies found evidence for negative cooperativity: both the run length and force increase when more kinesins carry cargo, but not in proportion to the increase in the number of kinesins (*Furuta et al., 2013*; *Jamison et al., 2010*). A few studies have shown that when multiple kinesins are on a cargo, only a fraction of them are the primary drivers (*Furuta et al., 2013*; *Jamison et al., 2010*). We find this to be true in our assay, with only 2-fold increase in the total number of kinesins driving and resisting the microtubule for every 16-fold increase in the kinesin surface concentration (*Figure 3—figure supplement 1*).

What causes this negative cooperativity? We find that it is due to a decrease in the run length and binding duration of driving and resisting kinesins as more kinesins participate in transport. *Figure 3F* shows that individual kinesin duration and run length decreases by 2-to-3-fold when kinesin surface concentration increases from 0.03 to 0.46 kinesin/µm² (~16 fold increase in surface concentration). (A decrease in both the run length and duration indicates that the kinesins have approximately the same velocity regardless of kinesin surface concentration, since velocity is displacement (run length) over time (duration)). This shorter duration and run length likely results from the higher tension between kinesins as more kinesins are transporting the cargo. Thus, using a force-gliding assay, we find that negative cooperativity of kinesins can be explained by the shorter run length of each participating kinesin.

## Multiple kinesins can rescue cargo motion by detaching other stuck kinesins from the microtubule

Motors need to walk in dense cellular environment and overcome myriads of roadblocks to transport the cargo to its destination (*Lakadamyali, 2014*). To dissect the mechanism of multiple kinesin-based transport in the presence of roadblocks, we introduced roadblocks in our force gliding assay. We labeled microtubules with commercially available quantum dots (~20 nm in size) using streptavidin-biotin linkage (*Sheung et al., 2018*). These are of a different color (QD605) than the QDs on kinesins (QD705) and could be separately detected. Consequently, we could track the motion of kinesins, microtubule, and roadblocks simultaneously.

Past studies found that a single kinesin either detaches immediately or pauses when it encounters a roadblock (*Schmidt et al., 2012*; *Schneider et al., 2015*). These are readily observable in the

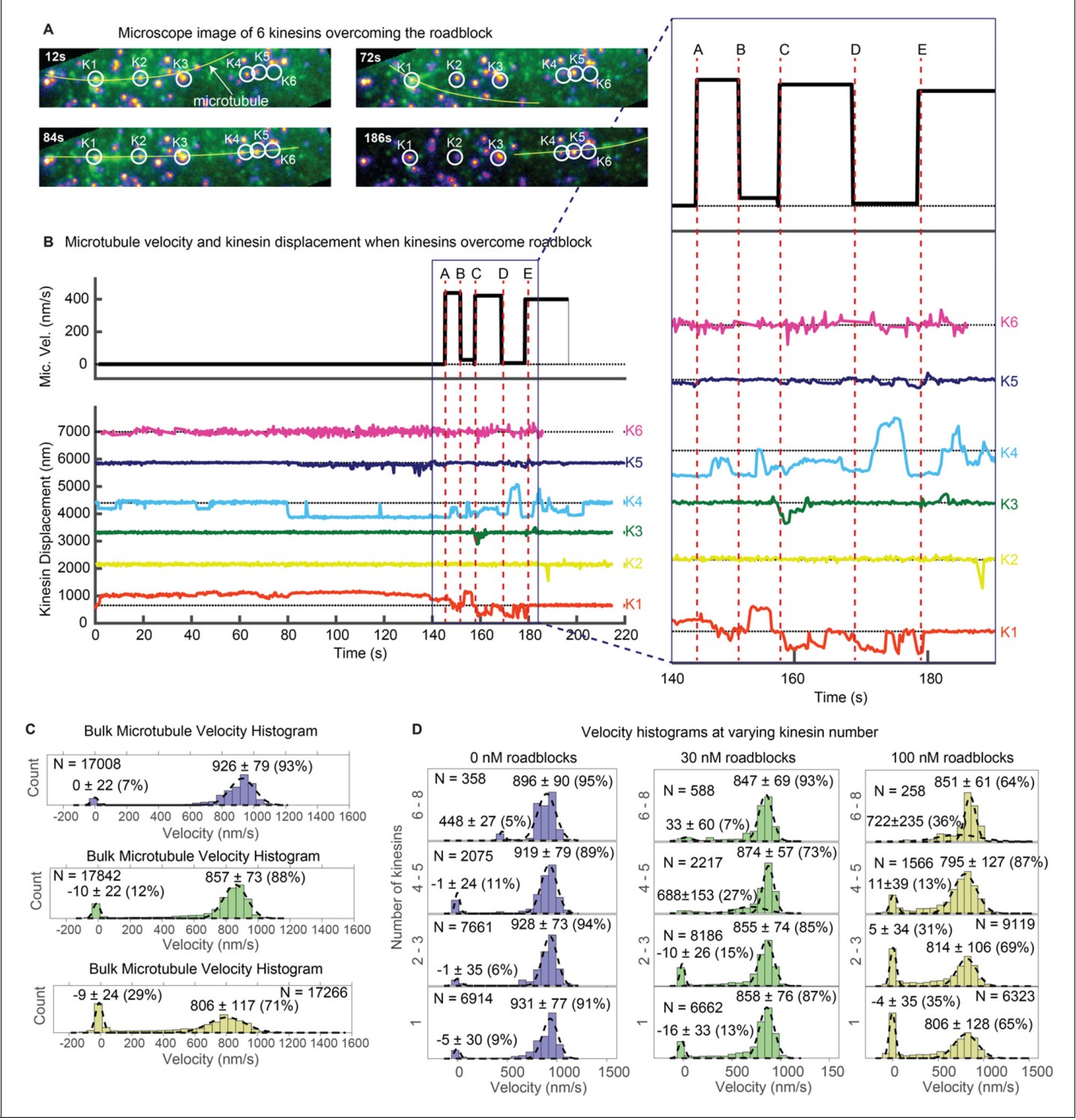

**Figure 4.** Multiple kinesins rescue stuck microtubule. (**A**) Images where six kinesins dynamically interact to rescue the motion of a stuck microtubule are shown (see **Video 5**). Kinesin-QD705s are shown as yellowish-orange spots, and microtubules are shown in green and overlaid with yellow lines for clear visualization. Four images at different time points are shown. Kinesin 1 (K1) is initially stuck at a roadblock (a QD605 attached to the microtubule) and microtubule is fluctuating around K1 (t = 12 s to t = 72 s). At t = 84 s, K4 catches the microtubule and makes failed attempts to drive the stuck microtubule. In the process, K4 stretches and straightens the microtubule, which becomes aligned with four other kinesins. More kinesins then start to interact with the microtubule and rescue its motion (image at t = 186 s). (**B**) Microtubule velocity and the displacements of six kinesin-QDs are plotted. Six time points (**A, B, C, D and E**) are marked by dotted red vertical lines, and zoomed in on the right. At first, the microtubule is stuck when K1 runs into a QD placed on the microtubule. The microtubule then diffuses around before catching onto K4 at ~ 80 s. K4 alone is not successful in detaching

*Figure 4 continued on next page*

*Figure 4 continued*

K1 from the roadblock, and only after K5 joins K4 in driving the microtubule at ~ 130 s does K1 start escaping from the roadblock. K1, K3, and K4 then alternately drive the microtubule. (C) Bulk microtubule velocity histogram, showing two distinct velocity populations: stuck (at 0 nm/s) and fast (at > 400 nm/s). As roadblock concentration increases from 0 nM (blue) to 30 nM (green) to 100 nM (yellow), the proportion of stuck microtubule increases from 7% to 29%, while the velocity of fast microtubules decreases from 926 nm/s to 806 nm/s. (D) With the force-gliding assay, we can further classify how many kinesins move the microtubule. At higher number of kinesins, the proportion of stuck microtubule decreases, while the velocity of fast microtubules remains approximately constant. Fits to double Gaussians are shown in dotted black lines.

The online version of this article includes the following figure supplement(s) for figure 4:

**Figure supplement 1.** Detachment of kinesin upon encounter with a roadblock.
**Figure supplement 2.** Kinesin detachment and pausing events upon roadblock encounter.
**Figure supplement 3.** Another example of microtubule rescue by the actions of multiple kinesins.
**Figure supplement 4.** Deriving mean kinesin velocity relative to microtubule.

force-gliding assay, as shown in *Figure 3—figure supplement 1*, *Figure 4—figure supplement 1*, and *Figure 4—figure supplement 2*, and *Video 2*, *Videos 3* and *4*. What is unclear is whether multiple kinesins can help cargo navigate through roadblocks. With the force-gliding assay, we can now observe multiple kinesins rescue a microtubule decorated with a QD as roadblock. The multiple kinesins do this by detaching from the microtubule a kinesin stuck at a roadblock, and then collectively guiding the microtubule forward.

*Figure 4* shows this. *Figure 4A* is a series of snapshots of *Video 5*. A microtubule decorated with roadblock (QD605) is depicted by a yellow line (marked with 'microtubule' at 12 s). The QD605 roadblock gets stuck at kinesin K1 (making it difficult to individually separate their fluorescence since they overlap) and eventually interacts with other kinesins (**K2–K6**) to rescue the microtubule motion. The six kinesin-QDs are shown in yellow-orange spots, enclosed in white circles. At t = 12 s to t = 72 s, the microtubule is fluctuating around K1, which is stuck at a roadblock on the microtubule. At t = 84 s, K4 catches the microtubule and makes failed attempts to drive the stuck microtubule. In the process, K4 stretches and straightens the microtubule, which becomes aligned with four other kinesins. More kinesins then start to interact with the microtubule and rescue its motion at t = 186 s.

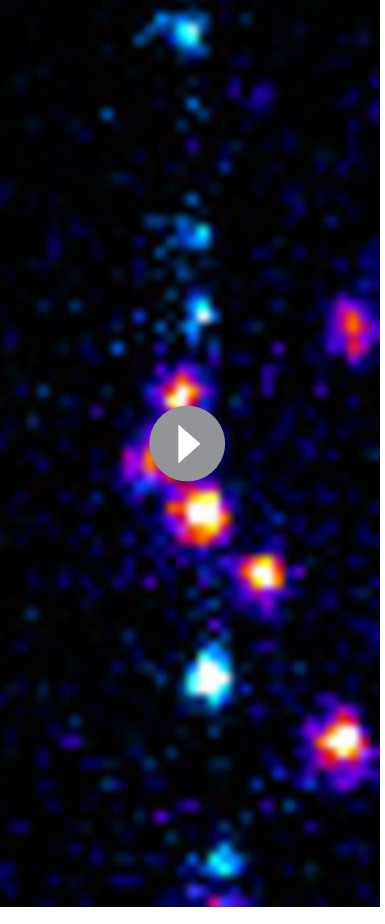

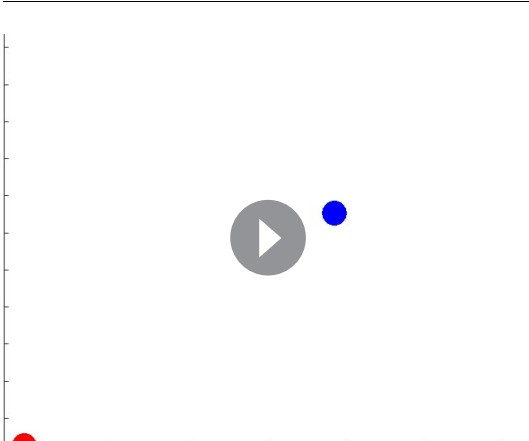

**Video 2.** Kinesin detaches upon roadblock encounter.
https://elifesciences.org/articles/50974#video2

**Video 3.** Kinesin detaches upon roadblock encounter - example 2.
https://elifesciences.org/articles/50974#video3

*Figure 4B* and its zoomed inset give a more detailed look. From 0 s to ~ 80 s, the QD605 road-block on the microtubule was stuck at K1. From ~ 80 s onwards, K4 attempted to drive the microtubule, but the microtubule did not budge until ~ 130 s, when K5 joined the drive attempt. K5 triggered a repositioning of the microtubule in the off-axis direction (see *Video 5*), and after a short lag, at ~ 143 s, the microtubule started moving (time A), driven by K4 with slight resistance from K1. At time B, the microtubule gets stuck again, because K1 starts to show greater resistance. This happens until time C, when K3 joins K4 in driving, and successfully propelling the microtubule forward. At time D, the microtubule stops again, now because of K4 resisting. This takes place up to time E, when K4 resumes driving. In summary, *Figure 4* shows that when a cargo (microtubule) is stuck at a roadblock (here a QD605 attached to the microtubule), multiple kinesins, by dynamically interacting with the microtubule, can rescue the motion of the cargo. After scanning through our entire dataset, we found 18 similar examples in which a microtubule is stuck at a kinesin and is rescued by the action of other kinesins. We show one additional example in Sup. *Figure 5*.

## Multiple kinesins ensure smooth transport in the presence of roadblocks

Next, we studied the statistics of kinesin motion in the presence of the roadblocks. Specifically, we varied the amount of roadblocks in the force gliding assay. In *Figure 4C*, we show the results of placing various concentrations of roadblocks (0, 30, and 100 nM streptavidin-QD) onto the biotinylated microtubule. Roadblock amounts of 0, 30, and 100 nM corresponded to linear densities ~ 0, 0.75, and 2.5 roadblock-QDs/micron of microtubule length. For simplicity, here we denote roadblock concentration in terms of nM. Upon plotting the velocity histograms of the microtubule, regardless of the number of kinesins (one through ~ 8), we find that the histograms are well represented by two Gaussian populations: stuck (~0 nm/s) and fast (800–950 nm/s). As the roadblock concentration increases from 0 to 30 to 100 nM, the proportion of stuck microtubule increases from 7% to 12% to 29%. The average velocity of the fast microtubules also decreases from 926 nm/sec to 857 nm/sec to 806 nm/sec, consistent with previous roadblock studies (*Chaudhary et al., 2018*). Taken together, roadblocks reduce the *average* cargo velocity and induce pauses in a cargo moved by a team of kinesin.

Because we use a gliding assay where we can determine the number of kinesins bound to the microtubule, we can further break down the bulk velocity histograms. We can estimate the variation in microtubule velocity with increasing number of kinesins transporting it (*Figure 4D*). When there are no roadblocks present (*Figure 4D* left column, 0 nM roadblocks), we observe that peak of the velocity remains almost constant as the number of kinesins increase. This result agrees with previous studies on multiple motors (*Derr et al., 2012*). On the other hand, the 100 nM roadblock data (*Figure 4D* right column) show a different result. Here, as the number of kinesin increases from 1 to 2–3 to 4–5, the proportion of stuck microtubules decreases from 35% to 31% to 13%, respectively, and eventually to zero when there are 6–8 kinesin moving the microtubule. The velocities of fast microtubules remain constant at around 805 nm/s regardless of motor number. This shows that roadblock-induced pauses can be reduced, eventually to nearly zero, by having more motors available to drive the cargo. One possible explanation of why having more motors makes the motion smoother is that forces of kinesins add up to induce higher tension on the stuck kinesin, which increases its detachment rate. Our results with optical trap studies in the next section further reinforces this explanation.

## Kinesin can collectively augment their force for overcoming the roadblocks

In support of the hypothesis that more motors help cargos overcome roadblocks and reduce cargo pauses, we observed nine instances when a resisting kinesin is stuck at a roadblock while other kinesins keep driving the microtubule, causing the stuck kinesin to detach from the surface. We present one of such case in *Figure 5*. As the resisting kinesin detaches from the coverslip, it starts moving with the microtubule, driven by four other driving kinesins (*Figure 5A* and *Video 6*). *Figure 5A* shows the images of the kinesin (yellow arrow) that detaches from the surface due to the force of four other driving kinesins (marked by white lines). At 64.8 s, four driving kinesins are pulling on a microtubule while one resisting kinesin is holding the microtubule back (*Figure 5A*). The resisting kinesin is then ripped

**Video 5.** Multiple kinesins help one resisting kinesin get unstuck from roadblock on microtubule.
https://elifesciences.org/articles/50974#video5

off of the glass coverslip and the kinesin travels

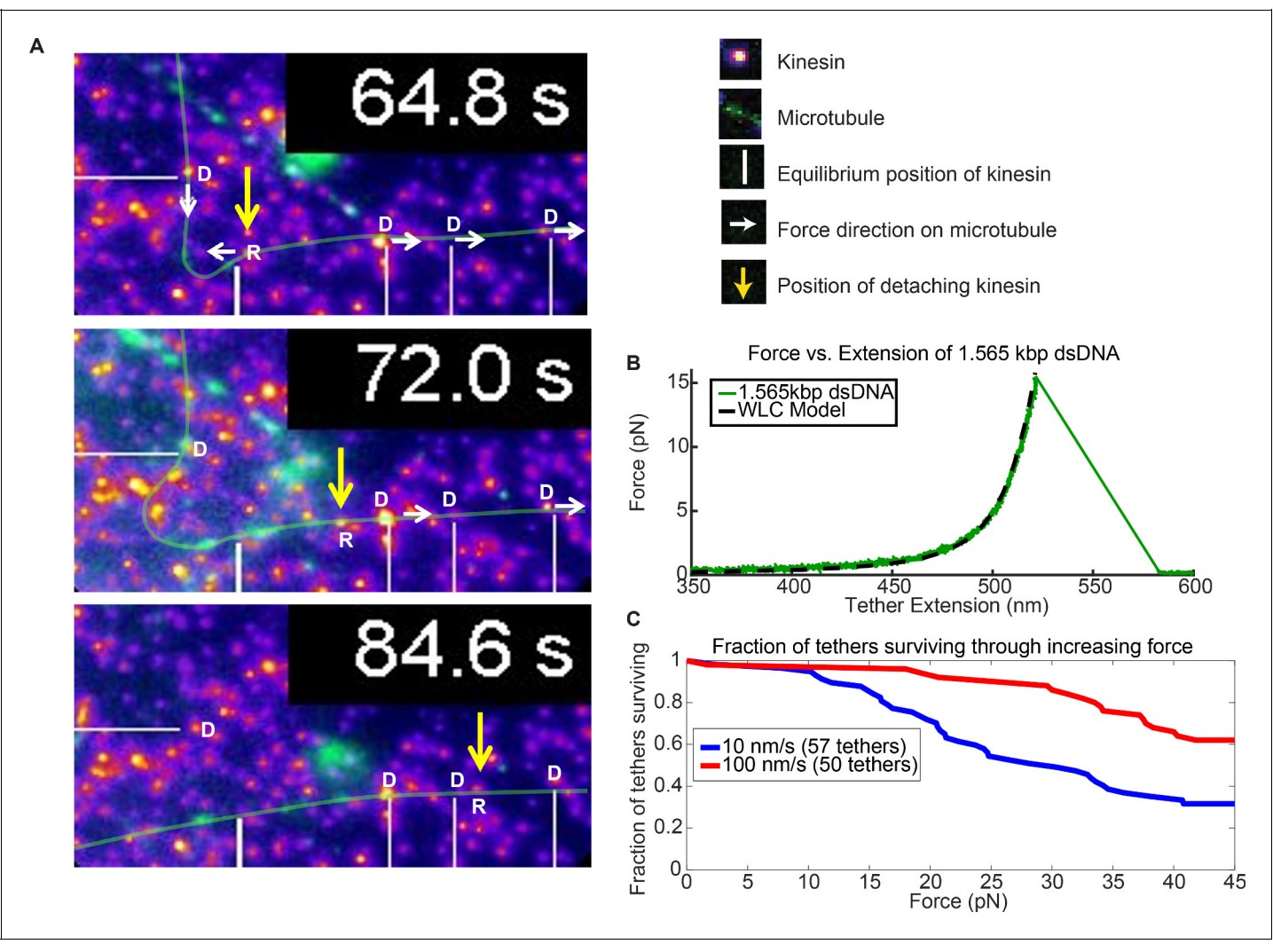

**Figure 5.** Force augmentation by kinesins. (**A**) Movie snapshots from *Video 6* showing 4 driving kinesins (depicted by letter D) and 1 resisting kinesin (depicted by letter R) moving a microtubule. At 64.8 s, the microtubule approached resisting kinesin but at 72.0 s and 84.6 s the resisting kinesin has broken free of its attachment to the coverslip by the combined forces of the 4 driving kinesins and is carried along with the microtubule. Yellow arrow shows the position the resisting kinesin that was detached from the coverslip. (**B**) Typical force extension curve of a single 1,565 kbp dsDNA obtained through a dual optical trap experiment. The digoxigenin:anti-digoxigenin bond is the weakest link holding the kinesin-DNA to the coverslip. The rupture force in this example is ~15 pN, when the DNA extension is ~520 nm. (**C**) Survival probability plot for rupture force for the DNA assembly. The force indirectly represents the force production by multiple motors.

The online version of this article includes the following figure supplement(s) for figure 5:

**Figure supplement 1.** Dual optical trap assay to measure the rupture force for digoxigenin:anti-digoxigenin interaction.
**Figure supplement 2.** Forced detachment traces and velocities.
**Figure supplement 3.** Tension range between multiple motors.
**Figure supplement 4.** Fraction of kinesin resisting from *Arpağ et al. (2014)*.

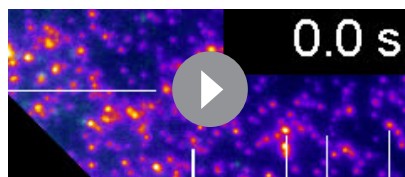

**Video 6.** Forced detachment of a resisting kinesin by multiple other kinesins.
https://elifesciences.org/articles/50974#video6

with the microtubule, as can be seen at 72.0 s and 84.6 s. Presumably, the weakest link—in this case, the digoxigenin:anti-digoxigenin antibody bond for the DNA linker, shown in the insert of *Figure 1C*, is rupturing (*Neuert et al., 2006*). This example shows that four kinesin are applying sufficient force to break the digoxigenin:anti-digoxigenin bond.

To test the magnitude of the force and compare it to the force that a single kinesin can exert (~6 pN), we then took the same 1.56 kb dsDNA and its linkages, and stretched it in an optical trap until the digoxigenin:anti-digoxigenin linkage ruptures (*Figure 5B*; *Figure 5—figure supplement 1*; Appendix 4). We pulled the dsDNAs at 10 nm/sec and 100 nm/sec loading rates (see *Figure 5—figure supplement 2* for rationale) and found half of the tethers ruptured at 30 and > 45 pN, respectively (*Figure 5C*; Appendix 5). These values are above the stall force of a single kinesin. Hence, the few (~4) driving kinesins observed pulling on the detaching kinesin may exert additive forces beyond what a single kinesin can exert. There is, however, an important caveat to this argument: due to the broad survival distribution, there is a small but non-negligible probability for the unbinding to occur at low forces, meaning the resisting kinesin could have released at < 6 pN (see Appendix 6).

## Discussion

We have dissected the dynamics of multiple kinesin-based cargo transport using our force-gliding assay. We can now observe individual kinesin dynamics and the effect of their dynamics on the instantaneous and overall cargo transport. We also studied cargo transport driven by multiple kinesins in the presence of roadblocks to mimic the cellular environment and to understand how multiple kinesins can overcome roadblocks.

### Dynamics and asymmetric response of kinesin during multiple motor transport

The force-gliding assay allows us to understand the interaction of individual kinesins with one another and with the microtubule. We found that underneath the seemingly smooth transport of a microtubule cargo, like that shown in *Figure 2B2*, multiple kinesins attached and detached frequently, switching their states from driving to resisting and vice versa (*Figure 2B3*). Thus, we uncover the hidden dynamics (i.e. the attachment-detachment, and the changing states) of kinesins previously unseen in simple motor walking or gliding assays.

Kinesin has two distinct states during cargo transport and these states have asymmetric run length and velocity response to load, depending on the load (or tension) direction. Driving kinesin dwells longer than resisting kinesin on the microtubule, and will slow down under hindering load caused by tension between kinesins. This tension is key in multiple motor transport, and our assay is the first experimental assay to show its presence between motors. Tension allows kinesins to communicate with one another: driving kinesin feels the tug of resisting kinesin and vice versa through this tension. This allows the driving kinesin to slow down and resisting kinesin to speed up, so that both driving and resisting kinesins walk in-sync at the same speed, that is the speed of the cargo.

Our assay also allows us to measure the force on individual kinesin motor. In our assay, we show that the tensions between kinesins vary between 0 to 4 pN. Since our assay uses a long and flexible linker (1565-base pair DNA), we therefore predict that in cells, where shorter and stiffer linkers (adaptor proteins) are employed, the tension will likely increase. This is because kinesin can travel further at low force with a long and flexible linker, and since there is a finite chance of dissociating at every step, kinesin can dissociate prematurely before high forces are reached when long and flexible linkers are used. This is confirmed by comparing our result with a simulation by *Arpağ et al. (2014)*, where shorter and stiffer linker (elasticity of 0.2 pN/nm after 40 nm stretching) is used. For a rough comparison, the DNA in our assay has an elasticity of 0.00066 pN/nm, 300-fold more elastic, at half the DNA contour length (i.e. 532/2 = 266 nm)—though the elasticity decreases at larger DNA extension (0.19 pN/nm at 516 nm). Arpağ et al. found that the tension in their simulation varies between 0

to 10 pN for driving kinesin and 0 to 15 pN for resisting kinesin (*Figure 5—figure supplement 3*), larger than the 0 to 4 pN tension measured in our system. Even though the tension in our system is likely lower than in the cell, measurements such as the fraction of kinesin resisting will likely remain the same. In fact, Arpağ et al. predicted that one-third of the driving/resisting kinesins will be resisting (*Figure 5—figure supplement 4*) (*Arpağ et al., 2014*), similar to the result of our measurement.

Inside cells, motors are bound to a lipid cargo, which may have varying fluidity. *Grover et al. (2016)* found that gliding velocity of microtubules transported by membrane-bound kinesin decreases with increasing membrane fluidity. This is due to the slippage of motor anchors in the lipid bilayer. Our studies are carried out with DNA tethers attached to a rigid, planar cargo (the coverslip surface), so cargo fluidity is not accounted for. Nevertheless, we predict that membrane fluidity will reduce the tension between the kinesins, though the ratio between driving and resisting kinesins will likely remain the same.

Another study on Myosin Va shows that fluid membrane allows vesicle travel at velocities up to twice that of a single motor (*Nelson et al., 2014*). This is due to the biased detachment of resisting motors. In fluid membrane, slippage of motor anchor causes resisting motors to lag behind the driving motor, and the detachment of this resisting, lagging motor will cause the vesicle to spring forward. Without a fluid membrane, resisting motors can detach when it is ahead or behind the driving motor, and the vesicle will spring backward or forward once detached (no biased detachment). Our study found that resisting kinesin motors will detach faster than driving motors, and we predict that, just like Myosin Va, a team of kinesin motors will travel faster on a fluid vesicle due to the biased detachment of resisting motors.

## Significance of a 1/3 resisting-kinesin and 2/3 driving-kinesin

The one-third resisting kinesin fraction (and two-third driving kinesin fraction) is a consequence of the asymmetric run length to load, since resisting kinesin detaches faster than driving kinesin at the same force. These resisting kinesins were not accounted for in published animations of kinesins working as a team (*Bolinsky et al., 2006*; *Condeelis et al., 2014*). What is the significance of a one-third (~33%) resisting kinesin? In particular, what happens when the fraction of resisting kinesin is 0% (no resisting kinesin) or 50% (equal fraction as driving)? We propose that the one-third resisting kinesin may be an optimal strategy to increase cargo run length and reduce tension between kinesins.

If there is no resisting kinesin (0%), any kinesin would have detached immediately as soon as it feels an assisting load. An immediate benefit is the absence of drag due to resisting kinesin. The downside is that there will be less kinesin attached to microtubule. Since more kinesin (even resisting ones) can help maintain attachment of cargo to microtubule. Less kinesin means that the cargo run length will be shorter. As a rough estimate, a 0% resisting kinesin strategy will reduce the run length a two-motor system from a two-motor run length of 8 μm (*Vershinin et al., 2007*) to a one-motor run length of 1 μm (*Vershinin et al., 2007*), assuming the resisting kinesin will detach at the slightest resisting force. Thus a 0% resisting kinesin strategy will reduce drag, but also reduce run length.

If there is an equal number of resisting kinesin as driving kinesin (50% resisting), the rate of resisting kinesin detaching will be the same as driving kinesin. If we have a kinesin with the resisting kinesin having the same detachment rate as the current driving kinesin, this would mean that the kinesin will stay resisting for a longer time before detaching, thus generating larger drag. The upside is that since the resisting kinesin can stay longer, the cargo run length will also be longer. Thus a 50% resisting kinesin strategy increase cargo run length, but also increases drag.

Since 0% resisting kinesin has low drag and short run length, while 50% resisting kinesin has high drag and long run length, we hypothesize that a 33% resisting kinesin strategy is a strategy that balances the drag and run length. It will be interesting to investigate this more thoroughly through future simulation studies.

## Despite negative cooperativity, force augmentation by multiple kinesins help cargo overcome roadblocks

We found that kinesin cooperates negatively in our assay: that is, when there are more available kinesins to bind the microtubule, only a fraction of them are actively driving or resisting at any one time. Past studies infer the net negative cooperativity of kinesin through stall forces of two kinesins (*Jamison et al., 2012*; *Jamison et al., 2010*). With force gliding assay, we are able to directly

observe that when kinesin surface concentration is increased 16-fold, there is only a 2-fold increase in the total number of kinesins driving and resisting the microtubule (*Figure 3—figure supplement 1*). Furthermore, we find that this negative cooperativity arise because the run length and binding duration of driving and resisting kinesins decrease as more kinesins participate in transport (*Figure 3F*).

Even though kinesins cooperate negatively, their forces can still combine additively on occasion (*Jamison et al., 2010*; *Vershinin et al., 2007*). In our assay, probing the effect of roadblocks on multiple motor cargo transport, we found that when one kinesin is stuck on a roadblock on a microtubule, other kinesins combine forces to help detach the stuck kinesin (*Figure 4A,B*). As a result, having more kinesins on the cargo leads to smoother cargo velocity and reduction of stuck cargo events (*Figure 4D*).

We also observed a limited number of cases where kinesin on the coverslip surface is detached and moved with roadblocks due to the combined forces of multiple kinesins (*Figure 5A*). These cases were more common at high roadblock concentration. We infer from such cases that the combined force was so high that it led to the detachment of the resisting kinesin from the surface. Using an optical trap, we indirectly quantified the force in the system and concluded that kinesins can augment their forces in the presence of roadblocks and, thus, can help in overcoming the roadblocks (*Figure 5B,C*).

## Model of multi-kinesin cargo transport

*Figure 6* is an example of how multiple kinesins might interact with a single cargo within a cell. When one kinesin moves slower than the cargo and becomes resisting (*Figure 6A*), the assisting forces from the cargo tend to increase this kinesin's speed or cause it to release rapidly, allowing the cargo to experience minimal drag force. Surprisingly, there appears to be ~ 35% resisting kinesins, causing a continuous tug-of-war among the kinesins, which tends to maintain an appreciable tension between kinesins. Kinesins can also rapidly switch between driving and resisting, leading to a fairly continuous and uninterrupted cargo motion forward. Against roadblocks, force augmentation of multiple kinesins may lead to large forces, causing detachment of resisting kinesin from cargo or microtubule (*Figure 6B*). By combining the motion and the force of single kinesins, we can connect the single molecule behavior of kinesins with their ensemble behavior. Whether the tension embedded within an in vivo system, and whether other molecular motors such as dynein and myosin have a similar cooperative behavior, remains to be seen.

# Materials and methods

## Key resources table

| Reagent type (species) or resource | Designation | Source or reference | Identifiers | Additional information |
|---|---|---|---|---|
| Peptide, recombinant protein | Truncated kinesin with 888 amino acids (K888) | Kathy Trybus lab (*Tjioe et al., 2018*) | | |
| Chemical compound, drug | Qdot 655 Streptavidin Conjugate | Thermo Fisher Scientific | Cat. # Q10123MP | |
| Chemical compound, drug | Qdot 705 Streptavidin Conjugate | Thermo Fisher Scientific | Cat. # Q10163MP | |
| Chemical compound, drug | THP | EMD Millipore | Cat. # 71194 | |
| Chemical compound, drug | Paclitaxel | Cytoskeleton, Inc | Cat. # TXD01 | |
| Chemical compound, drug | Adenosine 5'-triphosphate magnesium salt | Sigma Aldrich | Cat. # A9187 | |

*Continued on next page*

*Continued*

| Reagent type (species) or resource | Designation | Source or reference | Identifiers | Additional information |
|---|---|---|---|---|
| Chemical compound, drug | GMPCPP (Guanosine-5'-[(α,β)-methyleno]triphosphate, Sodium salt) | Jena Bioscience | Cat. # NU-405S | |
| Chemical compound, drug | AMP-PNP (Adenylyl-imidodiphosphate) | Sigma Aldrich | Cat. # 10102547001 | |
| Chemical compound, drug | Aminosilane (N-(2-Aminoethyl)—3-Aminopropyl trimethoxysilane) | United Chemical Technologies | Cat. # A0700 | |
| Chemical compound, drug | Biotin-PEG-Succinimidyl Valerate, MW 5,000 | Laysan Bio, Inc | Item# BIO-PEG-SVA-5K-100MG | |
| Chemical compound, drug | mPEG-Succinimidyl Valerate, MW 2,000 | Laysan Bio, Inc | Item# MPEG-SVA-2000–1 g | |
| Software, algorithm | Matlab code used for analysis | This paper | | Provided as *Source code 1*. |
| Software, algorithm | TrackMate in Fiji | ImageJ (https://imagej.net/Fiji/Downloads) | | |
| Software, algorithm | ImageJ (Fiji) | ImageJ (http://imagej.nih.gov/ij/) | | |
| Software, algorithm | Matlab | Matlab (https://www.mathworks.com/products/matlab.html) | | |

## Protein purification

Truncated kinesin with 888 amino acids (K888) from the mouse kinesin heavy chain (accession number BC090841) with a C-terminal biotin-tag and FLAG epitope, and mouse kinesin light chain (accession number BC014845) were cloned separately into the baculovirus transfer vector pAcSG2 (BD Biosciences) for recombinant virus production. Sf9 cells were infected with recombinant viruses, grown, harvested, lysed and purified using a published protocol for K888 homodimer kinesin (*Tjioe et al., 2018*). Briefly, infected cells in growth medium supplemented with 0.2 mg/ml biotin were harvested after 72 hr and lysed by sonication in lysis buffer (10 mM imidazole, pH 7.4, 0.3 M NaCl, 1 mM EGTA, 5 mM MgCl2, 7% (w/v) sucrose, 2 mM DTT, 0.5 mM 4-(2-aminoethyl) benzenesulfonyl fluoride, 5 µg/ml leupeptin) prior to clarifying at 200,000 x g for 40 min. The supernatant was applied to a FLAG-affinity column (Sigma-Aldrich) and washed with 10 mM imidazole, pH 7.4, 0.3 M NaCl, 1 mM EGTA. Specifically-bound protein was eluted in the same buffer containing 0.1 mg/ml FLAG peptide. Fractions of interest were combined, concentrated with an Amicon centrifugal filter device (Millipore), dialyzed against 10 mM imidazole, pH 7.4, 0.2 M NaCl, 1 mM tris(2-carboxyethyl)phosphine TCEP), 55% (v/v) glycerol,1 mM DTT, 1 µg/ml leupeptin, 50 µM MgATP, and flash frozen for storage at −80˚C.

## Magnetic cytoskeleton affinity (MiCA) purification of kinesin-QD

MiCA purification was performed to obtain one to one binding of biotinylated kinesin with streptavidin-QD 655 or 705. Briefly, kinesin K888 is mixed with 3x excess QD so that each QD has one or no kinesin bound 95% of the time. This reaction is allowed to incubate for > 10 min on ice in a BSA-taxol buffer (1 mM THP (71194, EMD Millipore), 20 µM Paclitaxel (Cytoskeleton, Inc) and ~30 nM ATP (Magnesium salt, A9187, Sigma Aldrich) in DmB-BSA (dynein motility buffer (30 mM HEPES, 50 mM KAcetate, 2 mM MgAcetate, 1 mM EGTA, pH 7.2) supplemented with 8 mg/mL BSA)) at 220 nM final K888 concentration and 660 nM final QD concentration. Excess QD is then removed through MiCA purification, which uses moderately positive magnetic beads (i.e. magnetic amine beads coated with PEG-amine to reduce highly positive amine charge) that bind to short

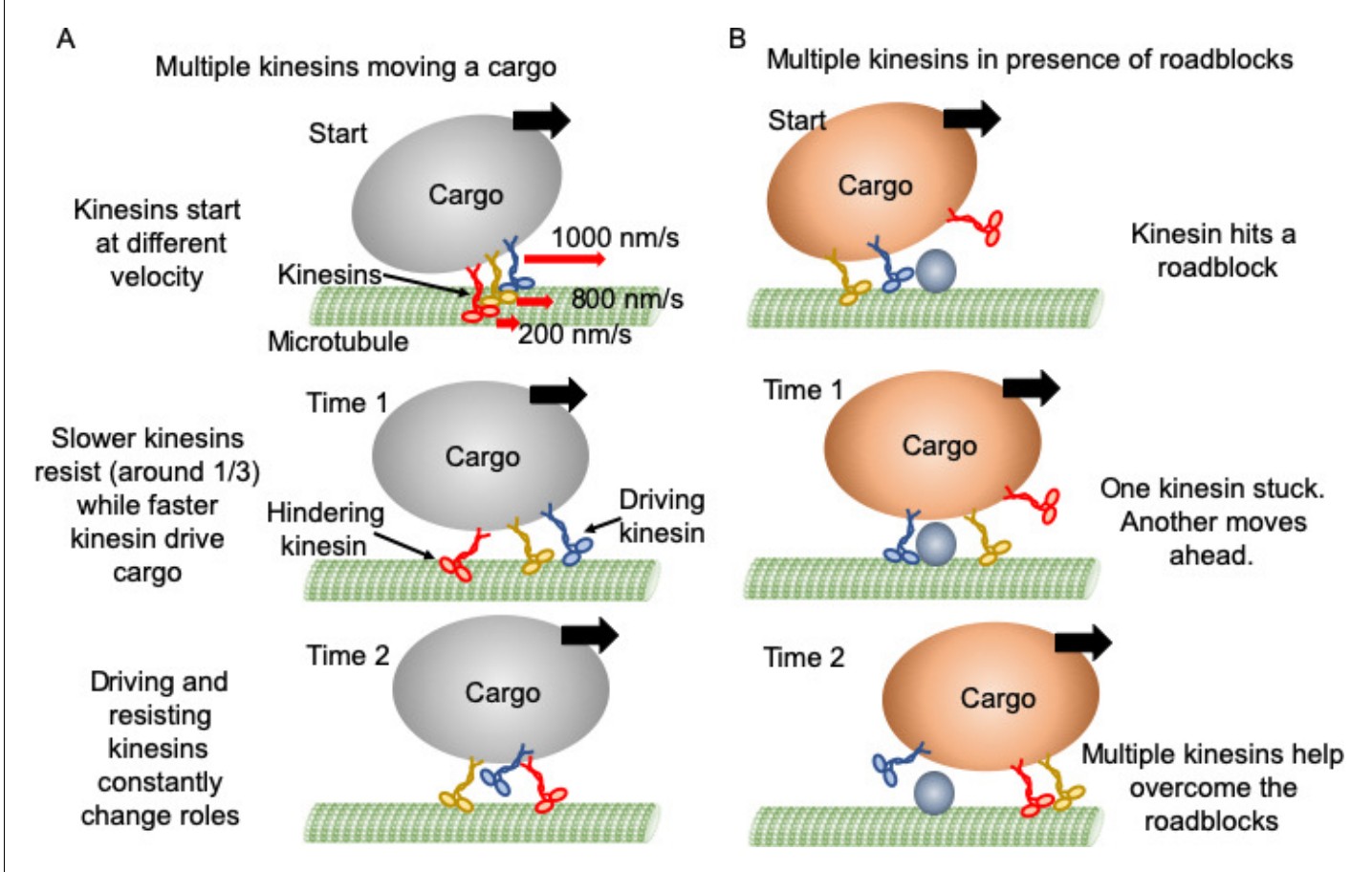

**Figure 6.** Mechanism of multiple kinesin based transport. (**A**) When working in a group, different kinesins may have different velocities while starting. Red arrows represent the initial velocity values of different kinesins. Thick black arrows represent the cargo velocity direction. At 'Time 1', faster kinesins will drive, while slower kinesins will resist the cargo motion. On average, one third of kinesins are resisting. Even though a resisting kinesin starts slower, eventually it reaches the same velocity as driving kinesins as it feels forward tension. At 'Time 2', kinesins candetach and switch dynamically between driving and resisting. Blue kinesin has detached from its driving position and red kinesin has shifted from resisting to driving position. Resisting kinesins spend a shorter duration on the microtubule before detaching compared to driving kinesins. Overall, a resisting kinesin presents little resistance to the forward motion when there is no roadblock, as it detaches 33% faster than driving kinesins. (**B**) In the presence of roadblocks, kinesins can get stuck at the roadblock (blue kinesin at 'Start' and 'Time 1'). One kinesin (purple at 'Time 1') may not generate enough tension to cause detachment of a resisting kinesin. However, additional kinesin(s) (red at 'Time 2') driving the cargo can generate enough force to rescue their stuck partner to resume cargo transport.

microtubules to form MiCA capture beads. This is done by mixing 5 µL sonicated GMPCPP microtubule (1 mg/mL short microtubules prepared from 97% pure tubulin (HTS03-A, Cytoskeleton, Inc), stored at −80˚C and thawed right before use) with 8 µL PEG-amine magnetic beads (10 mg/mL, prepared as previously published) with its buffer removed after a magnetic pull to leave only the pellet. After 5 min incubation in an end-to-end rotator at room temperature, the MiCA capture bead is washed 2x with 8 µL BSA-taxol buffer and reconstituted in 1 µL BSA-taxol buffer to give ~ 1.5 µL final bead volume. Next, 6 µL kinesin-QD (220 nM kinesin) is mixed with the 1.5 µL MiCA capture bead and 1.2 µL AMP-PNP (8 mM), and the mixture is allowed to incubate for 5 min at room temperature in an end-to-end rotator. The AMP-PNP causes kinesin-QD to bind strongly to MiCA capture beads. The mixture is then washed 3x with 8 µL BSA-taxol buffer and 8 µL elution buffer (2 mM ATP in BSA-taxol buffer) is added. After 5 min incubation in an end-to-end rotator at room temperature, the eluant is extracted, yielding approximately 80 nM kinesin-QD (assuming 50% purification yield).

## Force-gliding assay and roadblock experiment

22 square millimeter coverslips were sonicated in 1M KOH and plasma cleaned, then aminosilanized and reacted with N-hydroxysuccinimide (NHS) ester modified polyethylene glycol (PEG) that includes 1% biotin-PEG-NHS (*Roy et al., 2008*). The attachment of biotin-PEG to the surface is thus covalent. Double sided tape pieces were sandwiched between a thoroughly washed glass slide and the coverslip to make the imaging channels. 600 nM streptavidin was flowed into the channel and incubated for 5 min. The channel was washed with DMB-BSA buffer (30 mM HEPES, 50 mM KAcetate, 2 mM MgAcetate, 1 mM EGTA, 8 mg/ml BSA, pH 7.4). 10 nM biotinylated anti-digoxigenin (Abcam) was flowed into the chamber and incubated for 5 min followed by a subsequent wash with DMB-BSA buffer to remove excess anti-digoxigenin-biotin. MiCA purified kinesin-QD was mixed with eight times less DNA (IDT) to minimize conjugation of multiple DNA molecules to single kinesin-QD. The biotin end of DNA was conjugated with the kinesin-QD and the other end with digoxigenin remained free. Kinesin-QD-DNA was flowed into the chamber and the digoxigenin end of the DNA was conjugated with the Anti-digoxigenin on the surface. The chamber was incubated with excess biotin to saturate all the streptavidin binding sites in the chamber and subsequently washed with DMB-BSA. The number of kinesins on the surface were optimized such that they were sufficiently away from each other and could be tracked individually. Finally, the imaging buffer containing the polymerized microtubules, saturating ATP and deoxygenating agents (pyranose oxidase + glucose) was flowed in the imaging chamber and movies were acquired. Five sets of experiments were collected at 0.03, 0.06, 0.11, 0.23, and 0.46 kinesin/ $\mu m^2$. For each set, four to five movies (technical replicates) were imaged, each at 0.2 s exposure time for 1500 frames.

For doing the roadblock experiments, biotinylated-microtubules were incubated with equal volume of streptavidin-QD605 (Thermo Fisher Scientific) solution of varying concentration (0 nM, 30 nM, 100 nM QD605). Roadblock incubated microtubules were used in the imaging buffer for doing the roadblock experiments. Three sets of experiments were collected at no roadblock, 30 nM roadblock, and 100 nM roadblock (QD) concentration. For each set, four to five movies (technical replicates) were imaged, each at 0.2 s exposure time for 1500 frames.

## Rupture force experiment with optical tweezer

Double-stranded DNA was synthesized through PCR amplification of a 1.565-kbp segment of the pBR322 plasmid (New England Biolabs), using forward and reverse primers conjugated with a 5' biotin and a 5' digoxigenin, respectively (Integrated DNA Technologies) and a high-fidelity master mix (New England Biolabs). The PCR product was purified with a PCR cleanup kit (QIAGEN).

For optical trapping experiments, 2 or 2.4 μL of 0.05 nM dsDNA were incubated for an hour at room temperature with 5 μL of 0.2% w/v streptavidin-coated beads (Spherotech). Beads were diluted in approximately 300 μL of buffer (100 mM Tris, 20 mM NaCl, 3 mM MgCl$_2$, pH 7.6) for delivery to the optical traps through bead channels in a custom flow chamber (*Whitley et al., 2017*). In the trapping channel of the flow chamber, dual-trap optical tweezers were used to trap a DNA-coated streptavidin bead in one trap, and a bead (Spherotech) coated with digoxigenin (Roche Diagnostics) in the other. The beads were repeatedly brought together until a DNA tether formed.

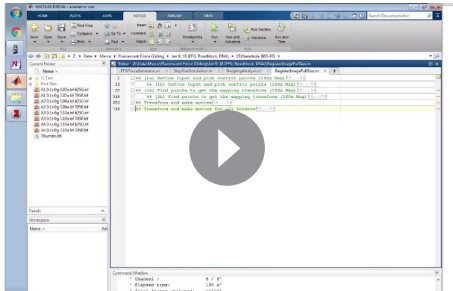

**Video 7.** Image registration tutorial.
https://elifesciences.org/articles/50974#video7

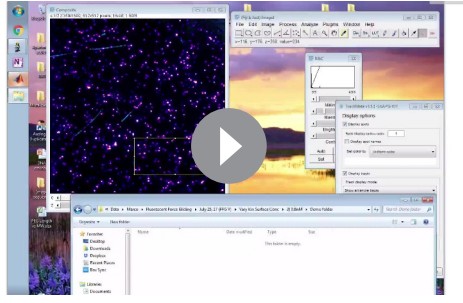

**Video 8.** Force-gliding analysis tutorial.
https://elifesciences.org/articles/50974#video8

Once a dsDNA tether was formed, a force-extension curve was collected by moving one trap away from the other at a constant rate (10 nm/s or 100 nm/s) over a pre-set distance, then returning at the same rate to the initial position. Most tethers ruptured during the force ramp. Rupture is expected to occur primarily at the linkage betwen digoxigenin and anti-digoxigenin, as rupture forces previously reported for this linkage (under different buffer conditions) have been lower than for the biotin-streptavidin linkage (*Merkel et al., 1999*; *Neuert et al., 2006*). Each resulting force-extension curve was fitted to the extensible worm-like chain model (*Camunas-Soler et al., 2016*; *Wang et al., 1997*) to verify that only one molecule was present and that it behaved correctly (*Figure 5B*). The maximum forces experienced by the single dsDNA tethers were determined and plotted as a survival distribution (*Figure 5C*).

The optical trapping experiments were conducted in a microfluidic flow chamber, in a channel containing trapping buffer consisting of 76% DmB-BSA (30 mM HEPES, 5 mM MgSO$_4$, 1 mM EGTA, pH 7.0 and 8 mg/ml BSA), 10 μM biotin, 100 μM ATP, 100 μM THP, 2 μM Paclitaxel, and an oxygen scavenging system (*Landry et al., 2009*; *Swoboda et al., 2012*) (final concentrations in buffer: 32 mg/mL glucose, 0.58 mg/mL catalase (from *Aspergillus niger*: Millipore Sigma, formerly EMD Millipore, 219261-100KU, 5668 U/mg), 1.16 mg/mL pyranose oxidase (from *Coriolus* sp.: Sigma P4234-250UN, 12.2 U/mg), 400 μM Tris-HCl and 2 mM NaCl).

## Image acquisition

Total Internal Reflection Fluorescence Microscopy (TIRFM) was performed with an inverted light microscope (Olympus IX71) equipped with two EMCCD cameras (iXon DU-897E), a TwinCam (Cairn Research) to split two colors into two separate cameras, a 100x magnification oil immersion objective (Olympus UPlanSApo, NA 1.40), and a green laser (10 mW power, Coherent OBIS 532 nm attenuated with a neutral density filter with optical density of 1.0. The excitation light was reflected with a 556 long-pass dichroic (T556lpxr-UF3 UltraFlat, Chroma) and cleaned up with 532 nm long-pass filter (BLP01-532R-25, Semrock). Fluorescence from QD and microtubule were split with a 685 nm long-pass filter (T685lpxr-UF3, UltraFlat, Chroma) in TwinCam. QD655, QD705 and a combined QD625 and HyLite 488 Microtubule emission were filtered using a 655/40 nm, 710/40, and 600/80 nm (BrightLine, Semrock) band-pass filter, respectively. Images were recorded with 0.2 s exposure time for all experiments, except for the experiment shown in *Figure 2*, where 0.1 s exposure time is used. An EM-gain between 10 and 300 was used, adjusted to maximize the signal collected without saturating the camera. No additional magnification was used for all experiments, except one shown in *Figure 2*, where 1.5x additional magnification is used. The pixel size for each image is thus 16,000 nm (the actual camera pixel dimension)/100 x objective magnification = 160 nm for most images, and 16,000 nm / 150 x total magnification = 106.7 nm for those with 1.5x additional magnification.

## Image registration and analysis

Fluorescent images obtained from the two channels of TwinCam were mapped onto each other using a transform file obtained from a set of nanohole images as previously described (*Tjioe et al., 2018*). The 512 × 512 pixels of combined image were visualized in Fiji (plugin-rich package of ImageJ) and gliding instances of every microtubule were cropped and saved. Point locations of all kinesin-QD were detected with TrackMate (*Tinevez et al., 2017*), a plugin within Fiji, using a Laplacian of Gaussian (LoG) detector, with estimated blob diameter of 4 pixels (160 nm/pixel), threshold of 50, and sub-pixel localization turned on. Simple LAP (Linear Assignment Problem) algorithm within TrackMate was used to track all detected spots, with maximum distance for frame-to-frame linking of 4 pixels, maximum distance for track segment gap closing of 4 pixels, and maximum frame gap of 20 frames. All spots detected and tracked were then saved as a csv file for subsequent analysis in Matlab. See *Video 7* for detailed tutorial.

In Matlab, kinesin-QD locations from TrackMate were imported, along with cropped images of microtubule and kinesin-QD. The Matlab code, FFGTraceGenerator.m, along with other necessary codes, are provided in Supplementary Material. Kinesin-QDs exhibiting driving and resisting were manually picked, and their on-axis displacements parallel to the microtubule axis were calculated after manual tracing of microtubule backbone using aggregated images from defined time-points (see *Video 8* from time 6:48 to 7:58). Microtubule bending is accounted for in the analysis. Variation in fluorescent intensity along a microtubule allows a microtubule kymograph to be generated. Edges

in the kymograph were detected using the 'edge' command in Matlab with the 'canny' detection method. Manual clean-up and patching of the edges were then done to make sure microtubule movements were captured for every frame. Next, all kymograph edges were converted into velocity and averaged to obtain the microtubule velocity over time. Microtubule displacement over time was then calculated from the velocity. See *Video 8* for detailed tutorial.

Kinesin-QD on and off-axis displacements along a microtubule were plotted and their equilibrium positions were manually identified. Drive and resist instances were then picked with the following criteria: 1) there must be at least two points with displacements more than 100 nm or larger than two standard deviations from the noise at equilibrium, and 2) traces with more than 5 s of missing data points are removed. All drive and resist instances were then saved, containing information such as the duration and kinesin-QD displacement over time. Microtubule length over time was then obtained by manually identifying the microtubule backbone at select frames.

Once all the drive and resist instances were identified for every cropped image, we compiled statistics including: the average kinesin drive-to-drive and drive-to-resist transitions; duration, run length, and force histograms; average kinesin velocity relative to microtubule over time; bulk microtubule velocity; and microtubule velocity for specific number of kinesins attached. Force was calculated from the kinesin-QD-DNA displacement by fitting an extensible Worm-Like-Chain (WLC) model with double stranded DNA contour length of 532 nm and persistence length of 50 nm. A distance offset of 20 nm was subtracted from the kinesin-QD-DNA displacement to account for the size of QD, proteins, and PEG and to arrive at the DNA extension length.

## Acknowledgements

This work was supported in part by NIH grants GM132392 and NSF PHY 1430124 (to PRS) and to NIH grant GM078097 (to KMT). We thank Michael Diehl for the helpful discussion on negative cooperativity. We would also like to thank Barbara Stekas for the help in preparing DNA leading to this study.

## Additional information

### Competing interests

Marco Tjioe: Marco Tjioe is affiliated with Element Biosciences. The author has no other competing interests to declare. The other authors declare that no competing interests exist.

### Funding

| Funder | Grant reference number | Author |
| --- | --- | --- |
| National Institutes of Health | GM132392 | Paul R Selvin |
| National Science Foundation | 1430124 | Paul R Selvin |
| National Institutes of Health | GM078097 | Kathleen M Trybus |

The funders had no role in study design, data collection and interpretation, or the decision to submit the work for publication.

### Author contributions

Marco Tjioe, Conceptualization, Data curation, Software, Formal analysis, Supervision, Validation, Investigation, Visualization, Methodology, Writing—original draft, Writing—review and editing; Saurabh Shukla, Rohit Vaidya, Alice Troitskaia, Software, Formal analysis, Visualization, Writing—original draft, Writing—review and editing; Carol S Bookwalter, Resources; Kathleen M Trybus, Resources, Supervision, Funding acquisition, Writing—review and editing; Yann R Chemla, Conceptualization, Software, Formal analysis, Supervision, Funding acquisition, Validation, Visualization, Writing—original draft, Writing—review and editing; Paul R Selvin, Conceptualization, Resources, Data curation, Formal analysis, Supervision, Funding acquisition, Validation, Investigation, Visualization, Methodology, Writing—original draft, Project administration, Writing—review and editing

## Author ORCIDs

Marco Tjioe (iD) https://orcid.org/0000-0002-6445-9828
Kathleen M Trybus (iD) http://orcid.org/0000-0002-5583-8500
Yann R Chemla (iD) http://orcid.org/0000-0001-9167-0234
Paul R Selvin (iD) https://orcid.org/0000-0002-3658-4218

## Decision letter and Author response

Decision letter https://doi.org/10.7554/eLife.50974.sa1
Author response https://doi.org/10.7554/eLife.50974.sa2

## Additional files

### Supplementary files

• Source code 1. Matlab code for force-gliding analysis.

• Transparent reporting form

### Data availability

All data generated or analysed during this study are included in the manuscript and supporting files.

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

## Appendix 1

For one-to-one labeling of kinesin to QD, we used an in house developed magnetic separation method, called MiCA purification (see Materials and methods) (*Tjioe et al., 2018*). The likelihood of having two or more DNAs bound to a QD is low due to the steric hindrance of bulky DNA (~230 nm end-to-end distance) bound to a small 20 nm QD, as well as the use of 8x less DNA to bind kinesin-QD during experiment (see Materials and methods). For our study, we draw our conclusions primarily on the basis of the relative numbers of driving and resisting kinesins. We define the relative numbers of kinesins as the number of driving kinesins in relation to the number of resisting kinesins (i.e. driving kinesin/resisting kinesin, or driving/resisting kinesin as a fraction of total kinesins). We cannot completely rule out non-fluorescent QDs (QD at off state due to blinking,~5% of the time at 0.2 s time frame; *Efros and Nesbitt, 2016*), which will underestimate the absolute number of kinesins measured (including those attached to fluorescent and non-fluorescent QD). Since the fraction of non-fluorescent QDs will be the same from experiment to experiment, the relative number of kinesins (e.g. between driving and resisting) and the trend comparing the number of kinesin across experiments would not be affected. Likewise, some driving and resisting events are missed due to small (<100 nm) kinesin-QD displacements (see Materials and methods), which will underestimate the absolute but not relative number and trend in the number of kinesins.

## Appendix 2

Theoretically, in our assay, if we can calculate the forces exerted by all the individual kinesins driving the microtubule, we can find the total force by just summing them up. However, in our assay this is not yet feasible. First, the forces have a magnitude as well as direction and must be vectorially added. Second, there is significant uncertainty in calculating the maximal force of any given kinesin because a small uncertainty in the displacement measurement gives rise to a large uncertainty in the corresponding force calculation. Consider the force-extension curve for the 1565 bp dsDNA (*Figure 5B*). The extensible worm-like chain model fits the force-extension curve very well; however, at an extension of 500 nm, for example, the slope of the DNA/WLC extension is ~ 1 nm/pN. Hence, a small displacement error, for example 10 nm would yield a possible error of 10 pN.

## Appendix 3

An interesting point in *Figure 3E* is that at about 1 s, the driving and resisting kinesin velocity actually crosses as they approach the microtubule velocity (~830 nm/s). This is due to the resisting and driving kinesin detaching or returning back to equilibrium. This return to equilibrium causes resisting kinesin to increase in velocity, and driving kinesin to decrease in velocity, as shown in *Figure 4—figure supplement 4b,d*, leading to the cross-over. This cross-over takes place at an earlier time for shorter kinesin traces and a later time for longer kinesin traces.

## Appendix 4

For the optical trap assay, once a tether is formed, one trap was moved away from the other at a constant rate (10 nm/s or 100 nm/s) over a pre-set distance, then allowed to return at the same rate to the initial position. For choosing the pulling rates for the optical trap experiment, we picked 9 instances of kinesin during forced detachment events. We calculated the kinesin velocity just before the forced detachment for all nine cases (See *Figure 4—figure supplement 3*). The kinesin velocity varied between 7 nm/s to 450 nm/s, with an average of 150 nm/s. We tested 10 nm/s and 100 nm/s pulling speed in the optical trap assay to find the lowest force needed to rupture the DNA (in general, the lower the pulling velocity, the lower is the force needed to rupture the digoxigenin:anti-digoxigenin interaction). More than 50% of tethers remain at the highest force pulled (45 pN), indicating that most of the digoxigenin: anti-digoxigenin linkage may take more than 45 pN to rupture.

## Appendix 5

Using the force to which half of the tethers survive gives a more accurate description of what the trapping experiments measure than using the average rupture force. This is because a considerable fraction of the tethers that were pulled did not rupture, but survived through a pulling and relaxing cycle. Given that the rupture value for digoxigenin:anti-digoxigenin linkage yields approximately 30 pN and 45 pN for 10 and 100 nm/s pulling rate, it is clear that the force exerted on the extracted kinesin (*Figure 4B,C*) must arise from more than one kinesin pulling on it. This shows that a group of kinesins driving a cargo can exert forces much greater than a single kinesin, and thus help in smooth cargo-transport despite the presence of roadblocks.

## Appendix 6

On the survival probability plot in *Figure 5C*, there is small proportion of digoxigenin:anti-digoxigenin bonds which rupture at 6 pN or less (~2–4%). We attempt to estimate the proportion of the nine detaching kinesins over the entire population, keeping in mind that there may be more detaching kinesins which we miss. We find that there are 221 resisting kinesins that do not detach, which have similar force conditions (DNA extension > 500 nm and loading rate of < 450 nm/s) as the nine detaching kinesins. This means that the nine detaching kinesins constitute 4% of the entire kinesin population, meaning there is a high probability that the detaching kinesins rupture at forces below 6 pN. Thus, even though we are confident that the detaching kinesin are pulled by multiple driving kinesins (on average 4), we are less certain that the forces these driving kinesins exert on the detaching kinesin are compounded beyond single kinesin stall force of 6 pN to break the digoxigenin:anti-digoxigenin bonds.

## Appendix 7

In *Figure 2B*, we observe the signature of velocity asymmetry in the plateaus of the kinesin displacement graphs. The plateau (shown as 2' region in *Figure 2B4 and B5*) means that the velocity of resisting kinesin (*Figure 2B4*) increases and driving kinesin (*Figure 2B5*) decreases to match the microtubule speed. How? Let's understand what happens before the plateau, that is at the 1' region? For resisting kinesin, the upward slope at 1' region means that kinesin's speed is slower than the microtubule speed (seen as an approximately linear slope). While this happens kinesin is moving further than its equilibrium position on the coverslip, and when the DNA connecting the kinesin and the coverslip gets stretched, the kinesin's speed is not allowed to lag behind microtubule's speed much further. At this point the kinesin is feeling a large assisting load pulling it forward, and it can choose to detach, or adapt its speed to the microtubule speed, which manifests as the plateau (2' region). At the plateau, kinesin has no displacement relative to the glass coverslip. Since the microtubule is moving at the same speed relative to both kinesin and glass coverslip, this means that at the plateau, kinesin's speed relative to microtubule matches the microtubule speed relative to the coverslip. We observe many cases of kinesins, both driving and resisting, matching their speeds to the cargo. In *Figure 2B* alone this happens to practically all of the driving and resisting instances. We find this surprising because without a force-gliding assay, we would only observe microtubule velocity (*Figure 2B1*), which fluctuates only mildly. With the force-gliding assay, we now understand that beneath the seemingly smooth microtubule movement there is considerable fluctuation in kinesin speed that matches and supports microtubule speed.

