## [Decision Letter]

**Acceptance summary:**

Your study provides new significant insights into unsolved multiple kinesin motor behavior. You developed a novel microtubule-gliding assay in which kinesin motors anchored to the coverslip surface via DNA tethers drive smooth gliding of microtubules. The ability to infer how motors communicate and engage in transport when operating collectively is important for understanding mechanisms of motor cooperation and potential interference. The tracking methodology and the analyses of multiple motor states (paused, and moving sub-states) are novel and highly informative. This represents a fundamentally new way to study single motor behavior when multiple motors carry a single cargo, and has led to the discovery of several interesting properties that are uniquely enabled by your technology.

**Decision letter after peer review:**

Thank you for submitting your article "Multiple kinesins induce tension for smooth cargo transport" for consideration by *eLife*. Your article has been reviewed by three peer reviewers, and the evaluation has been overseen by a Reviewing Editor and Suzanne Pfeffer as the Senior Editor. The following individuals involved in review of your submission have agreed to reveal their identity: Arne Gennerich (Reviewer #1); Masahide Kikkawa (Reviewer #2); Michael Diehl (Reviewer #3).

The reviewers have discussed the reviews with one another and the Reviewing Editor has drafted this decision to help you prepare a revised submission.

Summary:

In this manuscript, the authors provide new significant insights into unsolved multiple kinesin motor behavior. When considering cargo transport in the cell, it is necessary to describe the behavior of multiple motors. However, it has been challenging to observe individual motors while they are transporting one cargo together.

The authors demonstrate the application of an in vitro microtubule-gliding assay in which kinesin motors anchored to the coverslip surface via DNA tethers drive smooth microtubule gliding. To determine whether the microtubule-attached kinesin motors drive or resist microtubule transport, the authors track the quantum dot-tagged kinesin motors with nanometer-scale precision. In addition, by calculating the extensions of the DNA tethers of the individual microtubule-bound and coverslip-attached kinesins, the authors estimate the forces that the kinesin motors experience during the active gliding of microtubules using a worm-like chain model. The authors find that multiple microtubule-attached kinesin motors exert tension on each other (up to 4 pN) and that, on average, one-third of the motors resist microtubule gliding, while two-thirds drive microtubule motion. The authors further show that the resisting motors detach more rapidly from the microtubule than the actively driving motors and the driving motors can speed up the resisting motors. Furthermore, the authors demonstrate that multiple kinesin motors acting on the same microtubule can help to overcome roadblock-induced pausing and thereby facilitate smooth microtubule gliding. The ability to infer how motors communicate and engage in transport when operating collectively is important to understanding mechanisms of motor cooperation and potential interference. The tracking methodology and the analyses of multiple motor states (paused, and moving sub-states) are comprehensive and sound. The study will be of broad interest to the cytoskeletal motor community. This represents a fundamentally new way to study single motor behavior when multiple motors carry a single cargo, and has led to the discovery of several interesting properties that are uniquely enabled by their technology.

Essential revisions:

1) While the authors attach the kinesin motors to the cover slip surface via DNA tethers, the DNA tethers are attached to a rigid "cargo" (the cover slip surface). It is therefore not clear whether the reported behaviors of the groups of kinesin motors that transport microtubules along the cover slip surface will be similar or different to the behaviors of the kinesin motors that are bound to a fluid lipid membrane. The authors should at least discuss the differences between the employed microtubule-gliding assay and the in vivo transport of membranous cargo. We recommend discussing and citing the recent work by the Diez lab (Grover et al., 2016) that has shown that the efficiency of transport of membrane-anchored kinesin depends on the diffusivity of the motors.

2) Figure 3, figure legend. The authors write that "Driving kinesins start at a higher relative velocity (~1300 nm/s) than resisting kinesins (~500 nm/s). This is due to natural variation in kinesin velocity". It is not clear why the authors think that the natural (Poissonian) variation in kinesin velocity explains why the resisting kinesins move on average at a lower velocity than the driving kinesins. Isn't it more likely that the resisting kinesins move slower as they are "forced" to move forward against their slow or resisting motion while the driving motors move faster as they form the dominating active group (2/3 vs. 1/3)? Please clarify.

3) Subsection “Kinesin can collectively augment their force for overcoming the roadblocks”: The authors write that the forces generated by a group of kinesin motors can break the digoxigenin:anti-digoxigenin bond and measure the strength of this bond using optical tweezers. From the provided methods, it is not clear whether the PEG coating is covalently bound to the cover glass as no references are provided (if they used the methods established by TJ Ha's group, the authors should say so). It is therefore not clear whether the digoxigenin:anti-digoxigenin bond breaks or whether the PEG with the attached anti-digoxigenin antibody detaches from the glass. The authors should discuss this possibility and update their Materials and methods section.

---

## [Author Response]

Essential revisions:1) While the authors attach the kinesin motors to the cover slip surface via DNA tethers, the DNA tethers are attached to a rigid "cargo" (the cover slip surface). It is therefore not clear whether the reported behaviors of the groups of kinesin motors that transport microtubules along the cover slip surface will be similar or different to the behaviors of the kinesin motors that are bound to a fluid lipid membrane. The authors should at least discuss the differences between the employed microtubule-gliding assay and the in vivo transport of membranous cargo. We recommend discussing and citing the recent work by the Diez lab (Grover et al., 2016) that has shown that the efficiency of transport of membrane-anchored kinesin depends on the diffusivity of the motors.

This is a good discussion to have. We have added the following paragraphs:

“Inside cells, motors are bound to a lipid cargo, which may have varying fluidity. Grover et al. found that gliding velocity of microtubules transported by membrane-bound kinesin decreases with increasing membrane fluidity (Grover et al., 2016). […] Our study found that resisting kinesin motors will detach faster than driving motors, and we predict that, just like Myosin Va, a team of kinesin motors will travel faster on a fluid vesicle due to the biased detachment of resisting motors.”

2) Figure 3, figure legend. The authors write that "Driving kinesins start at a higher relative velocity (~1300 nm/s) than resisting kinesins (~500 nm/s). This is due to natural variation in kinesin velocity". It is not clear why the authors think that the natural (Poissonian) variation in kinesin velocity explains why the resisting kinesins move on average at a lower velocity than the driving kinesins. Isn't it more likely that the resisting kinesins move slower as they are "forced" to move forward against their slow or resisting motion while the driving motors move faster as they form the dominating active group (2/3 vs. 1/3)? Please clarify.

We say that this is a natural (Poissonian) variation because at the start, the kinesins do not feel any force (or tension) from other kinesins since the DNA is not stretched. So the kinesin initially simply moves as it would when it is carrying the cargo by itself. And here there are distributions of velocity, some move faster than the others. When they are connected to the same cargo (in this case the large coverslip), the faster ones get classified as driving (since it moves faster than the microtubule velocity and got displaced to the plus end of microtubule), and slower ones get classified as hindering (as it got displaced in the opposite direction, to the minus end of the microtubule). Those that move at average velocity likely will have similar velocity as microtubule at the start, and will not be classified as driving or hindering, until their velocity differ from that of the microtubule.

We have made the following changes to the main text to help clarify this:

“Driving kinesins start at a higher relative velocity (~1300 nm/s) than resisting kinesins (~500 nm/s). This is due to natural variation in kinesin velocity, as at the start of motion kinesin feels negligible force from other kinesins (DNA is not stretched).”

3) Subsection “Kinesin can collectively augment their force for overcoming the roadblocks”: The authors write that the forces generated by a group of kinesin motors can break the digoxigenin:anti-digoxigenin bond and measure the strength of this bond using optical tweezers. From the provided methods, it is not clear whether the PEG coating is covalently bound to the cover glass as no references are provided (if they used the methods established by TJ Ha's group, the authors should say so). It is therefore not clear whether the digoxigenin:anti-digoxigenin bond breaks or whether the PEG with the attached anti-digoxigenin antibody detaches from the glass. The authors should discuss this possibility and update their Materials and methods section.

The PEG coating is covalently bound to the cover glass. We have updated the Materials and methods section as follows:

“22 square millimeter coverslips were sonicated in 1M KOH and plasma cleaned, then aminosilanized and reacted with N-hydroxysuccinimide (NHS) ester modified polyethylene glycol (PEG) that includes 1% biotin-PEG-NHS (Roy, Hohng, and Ha, 2008)”.